# SCALE: Augmenting Content Analysis via LLM Agents and AI-Human Collaboration

## Abstract

Content analysis is a fundamental social science research method that breaks down complex, unstructured texts into theory-informed numerical categories. It has been widely applied across social science disciplines such as political science, media and communication, sociology, and psychology for over a century. This process often relies on multiple rounds of manual annotation and discussion. While rigorous, content analysis is domain knowledge-dependent, labor-intensive, and time-consuming, posing challenges of subjectivity and scalability. In this paper, we introduce SCALE, a transformative multi-agent framework to **S**imulate **C**ontent **A**nalysis via large language model (**LL**M) ag**E**nts. This framework automates key phases including text coding, inter-agent discussion, and dynamic codebook updating, capturing human researchers' reflective depth and adaptive discussions. It also incorporates human intervention, enabling different modes of AI-human expert collaboration to mitigate algorithmic bias and enhance contextual sensitivity. Extensive evaluations across real-world datasets demonstrate that SCALE exhibits versatility across diverse contexts and approximates human judgment in complex annotation tasks commonly required for content analysis. Our findings have the potential to transform social science and machine learning by demonstrating how an appropriately designed multi-agent system can automate complex, domain-expert-dependent interactions and generate large-scale, quality outputs invaluable for social scientists.

## 1 Introduction

Content analysis is a cornerstone research method in the social sciences and humanities, offering a systematic and quantitative approach to interpreting complex, unstructured data (Holsti, 1969; Krippendorff, 2018; Neuendorf, 2017; Riffe et al., 2023). It converts qualitative information into structured, quantitative data by categorizing text based on theory-driven frameworks (Krippendorff, 2018; Weber, 1990) for scholars across disciplines including political science (Benoit, 2014), sociology (Dart, 2014), communication (Macnamara, 2005), and psychology (Hara et al., 2000).

However, traditional content analysis is labor-intensive and time-consuming (Hopkins & King, 2010; Zhao & Wong, 2024). Its standard procedures require a team of researchers to manually annotate sizable datasets (e.g., 500–1,000 entries), iteratively refining coding schemes and rules in 3–5 rounds to ensure reliability and validity of findings (Cohen, 1960; Krippendorff, 2018; Riffe et al., 2023). This manual process, while rigorous, presents two challenges: First, it relies heavily on domain-specific knowledge and individual scholars, potentially introducing subjectivity and limiting generalizability Second, the need for substantial human resources makes it difficult to scale, particularly as the volume of digital data continues to grow exponentially.

Recent advancements in artificial intelligence (AI), particularly in the development of large language models (LLMs), present opportunities to address these challenges (Ziems et al., 2024). LLMs have demonstrated remarkable capabilities in natural language understanding and generation (Zhao et al., 2023; Achiam et al., 2023; Kevian et al., 2024; Team et al., 2023), offering a potential solution to automate the content analysis process. However, existing LLM-driven approaches often lack the depth of human-like reasoning and adaptability, limiting their effectiveness in domain-specific tasks that require fine-grained understanding and iterative refinement.

In this paper, we propose a novel multi-agent framework to **S**imulate **C**ontent **A**nalysis via **LL**M ag**E**nts (SCALE), as shown in Figure 1. Our framework introduces a transformative approach by automating key phases of content analysis, including text coding, inter-agent discussion, and dynamic codebook evolution. Unlike previous methods, SCALE is designed to capture the reflective depth and adaptive discussions characteristic of human researchers, thereby reducing subjectivity and improving scalability. Moreover, by incorporating different human–AI collaboration modes inspired by social influence theories (Cialdini & Cialdini, 2007; French, 1959) and human-computer interaction theories (Suchman, 1987; Sundar, 2020), our framework augments multi-agent interactions with human expert intervention. This potentially mitigates algorithmic bias and strengthens contextual sensitivity, making it suitable for a wide range of social science content analysis tasks.

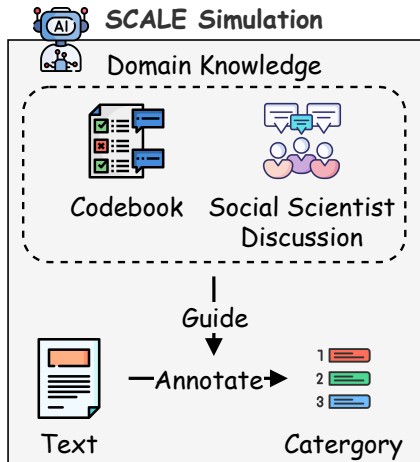

Figure 1: Illustration of augmented content analysis. Our multi-agent framework, SCALE, is proposed to tackle this complicated reasoning task by automating text coding, inter-agent discussion, and dynamic codebook evolution.

We evaluate SCALE on multiple real-world datasets, demonstrating its versatility across diverse contexts and its ability to approximate human judgment in complex annotation tasks. Developed in collaboration with social scientists, we demonstrate the potential of our framework to revolutionize content analysis in the social sciences and humanities, providing researchers with a scalable, efficient, and reliable tool for analyzing large-scale textual data. Our work's contributions are fourfold.

* **Scalability Enabler.** By harnessing the generative power of LLM, our proposed SCALE significantly reduces the time, human resources, and costs traditionally required for content analysis, enabling large-scale, high-quality annotation of complex content that was previously infeasible in social science. To the best of our knowledge, this is the first LLM work to capture and simulate the rigorous and dynamic process of quantitative content analysis.

* **Use-Inspired Design.** SCALE's design incorporates the domain knowledge of social science through the deep involvement of a social scientist. Its key phases—independent text coding, inter-agent discussions, and dynamic codebook updates—faithfully reflect the principles and standards of manual content analysis while being implemented within a computing framework.

* **Human Intervention;** Our framework provides a user-friendly interface for domain experts to intervene at custom modes and levels. By incorporating expert input—whether as a persuader or supervisor in human–AI interactions—this theory-informed integration augments AI decision-making and mitigates LLM bias.

* **Extensive Validation.** SCALE demonstrates effectiveness across content analysis tasks involving diverse topics. Our comprehensive experimental evaluations and analyses by domain experts confirm that SCALE can closely mimic human judgment in content analysis, delivering automated, valid, and reliable results invaluable for large-scale social science tasks.

* **AI for Social Good.** By simulating content analysis through multiple LLM agents, our framework empowers diverse social science and humanity communities that traditionally rely on manual methods. This allows them to conduct large-scale research with substantial societal impact and develop AI-powered solutions for pressing, real-world issues, potentially accelerating socially significant research and contributing to AI for social good.

## 2 RELATED WORKS

**Content Analysis.** Content analysis has long been a foundational method in the social sciences and humanities, providing a structured approach to converting qualitative text into quantitative data (Holsti, 1969; Krippendorff, 2018; Neuendorf, 2017; Riffe et al., 2023). Traditional content analysis methods have been applied across disciplines like political science (Benoit, 2014), sociology (Dart, 2014), media studies (Macnamara, 2005), and psychology (Hara et al., 2000). Recently, content

analysis has significantly advanced the understanding of complex social issues, ranging from political polarization (Conover et al., 2011) to emotional contagion (Kramer et al., 2014) and group dynamics (Holsti, 1969). These methods rely on manual annotation by human coders, who use predefined rules to categorize text, often iteratively refining their coding schemes in multiple rounds of discussions (Riffe et al., 2023). Although manual content analysis provides robust, theory-driven insights, it remains labor-intensive, time-consuming, and prone to subjectivity (Hopkins & King, 2010). Furthermore, as the volume of digital text increases, scaling traditional methods to accommodate larger datasets has become increasingly challenging (Zhao & Wong, 2024).

Recent advances in AI, particularly in natural language processing (NLP) and large language models (LLMs), are beginning to offer automated solutions for content analysis (Eloundou et al., 2023; Achiam et al., 2023; Tan et al., 2024). Automated content analysis using machine learning techniques can efficiently analyze large-scale datasets while maintaining accuracy in text categorization (Chew et al., 2023; Ziems et al., 2024). However, these methods still struggle to match the nuanced judgment of human experts, especially in subject domains where context and interpretative depth are crucial (Team et al., 2023). Therefore, an urgent need exists for advanced frameworks that integrate AI's scalability with the depth and adaptability of human judgment.

**Multi-agent Systems for Social Science.** Multi-agent systems (MAS) have become increasingly prevalent in computational social science, modeling social phenomena through agent interactions representing individuals or groups with predefined behaviors or decision-making rules. (Van der Hoek & Wooldridge, 2008; Chen et al., 2021; Chmura & Pitz, 2007; Macal, 2016; Lee et al., 2018; Chen et al., 2018; Dehkordi et al., 2023). Recent work explores MAS by simulating human-like deliberation for more nuanced decision-making such as data interpretation (Gürcan, 2024; Turgut & Bozdag, 2023). However, existing systems often lack mechanisms of inter-agent interactions or dynamic updates of decision rules (Gheyle & Jacobs, 2017). To fill this gap, our framework innovatively integrates LLM-based agents to simulate independent human coder deliberation, faciliate iterative, adaptive discussions between coders, and allow for dynamic updates of coding rules.

**Human Intervention.** Human intervention remains essential for the reliable deployment of AI-driven systems (Renner, 2020; Shoshitaishvili et al., 2017). As a general framework, Human-in-the-loop (HITL) systems allow experts to refine AI outputs, ensuring alignment with domain-specific knowledge and mitigating algorithmic bias (Mosqueira-Rey et al., 2023; Ghai & Mueller, 2022; Xu et al., 2023; Jolfaei et al., 2022). This is particularly important in social sciences and humanities, where interpretative depth and contextual sensitivity are critical (Goodsell, 2013). Recent approaches (Arambepola & Munasinghe, 2021) integrate expert feedback to adjust categories or coding schemes iteratively. Our framework significantly extends this line of work by designing different modes of human–AI collaboration informed by social influence theories (Cialdini & Cialdini, 2007; French, 1959) and human-computer interaction theories (Suchman, 1987; Sundar, 2020). First, AI and humans can collaborate through two relational dynamics, depending on the level of authority in their roles: persuasion (a collaborative structure) or supervision (a top-down structure). While persuasion may foster a more mutual, collaborative learning loop, supervision tends to be more straightforward and efficient. Second, human–AI collaboration can differ in terms of how much control humans have over AI outputs. Higher human control spanning all phases of content analysis increases costs and reduces automation, but it enables AI to better align their reasoning with human experts over time. A theory-informed factorial experiment with four conditions, combining the two factors, enables us to identify the most effective and efficient mode of human intervention.

## 3 PRELIMINARY: CONTENT ANALYSIS IN SOCIAL SCIENCE

Social scientists conduct *content analysis* by manually annotating textual data to uncover potential patterns and insights. Two or more social scientists first develop a codebook with a set of coding rules, grounded in relevant social science theories and contextualized within the given text corpus. Guided by the codebook, each social scientist then independently labels a small set of text entries (e.g., 10–20), after which they meet to discuss and resolve inconsistencies, leading to more refined and specific coding rules in the updated codebook. This process iterates for 3–5 rounds until convergence. The finalized codebook is applied by each social scientist to separately label a larger number of different text entries. Despite its rigor, content analysis is highly labor-intensive, time-consuming, and subject to individual biases, which presents challenges in terms of scalability and consistency.

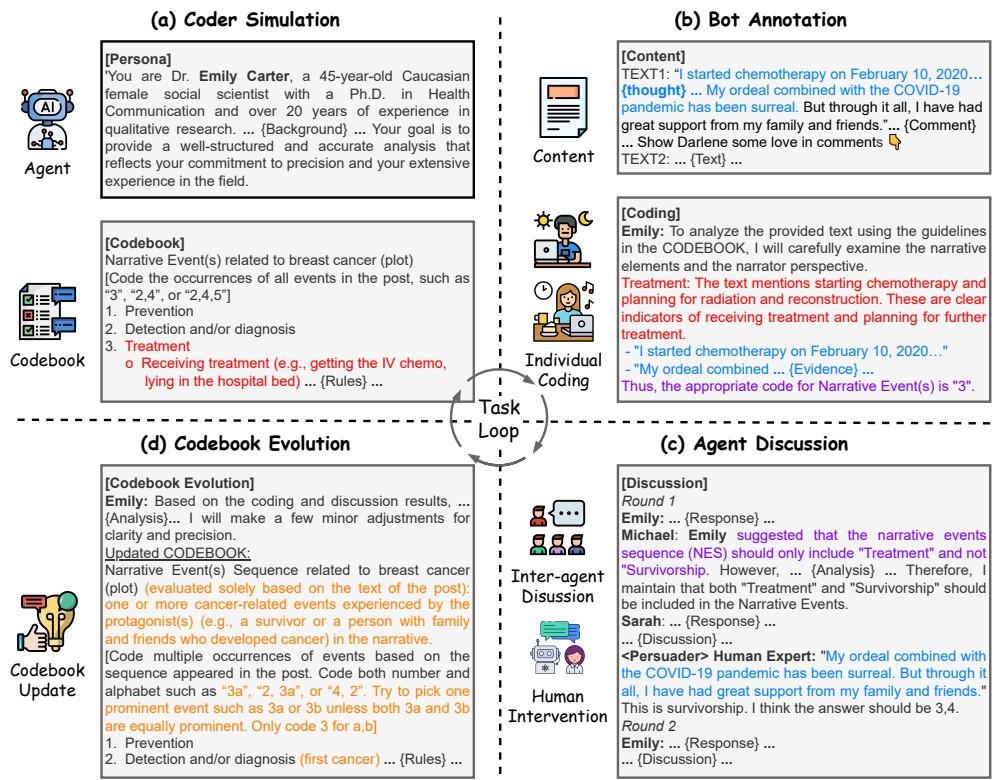

Figure 2: Proposed SCALE framework. (a) Coder Simulation. Initialize LLM agents and the codebook with real-world persona and predefined rules, respectively. (b) Bot Annotation. LLM agents independently code a batch of text into theory-informed categories following the codebook (c) Discussion. LLM agents conduct multi-round discussions to negotiate inconsistent results. Optional human interventions can be involved here or (in large scope) to control agents' behavior. (d) Codebook Evolution. Based on the results of the coding and discussion, LLM agents will update and discuss the update of the codebook. The final version will be used in the next-round workflow. Red text: codebook. Blue text: text. Purple text: discussion. Orange text: evolution.

## 4 METHODOLOGY

### 4.1 PROPOSED SCALE FOR CONTENT ANALYSIS

We propose SCALE to **S**imulate **C**ontent **A**nalysis via **L**LM multiag**E**nt by automating key phases, including text coding, inter-agent discussion, and dynamic codebook updating. Overall, our method can be illustrated in four steps as illustrated in Figure 2.

**Coder Simulation.** Before the simulation, we initialize agents and the codebook. We first initialize $N$ LLM agents $\mathcal{A} = \{a_i\}_{i=1}^N$ to enact well-trained social scientists using system prompts with $N$ persona $\mathcal{P} = \{p_i\}_{i=1}^N$, which, except names, is derived from the real world social scientist for authentic role-playing. Based on the specific task, we initiate a codebook $\mathcal{C}$ with $N_r$ human-expert predefined rules $\mathcal{C} = \{r_i\}_{i=1}^{N_r}$ or empty set $\varnothing$ where agents need to propose and update the codebook from scratch. For simplicity, we consider each rule to represent one scenario and can be used to code text into one unique discrete category. As shown in Figure 2(a), one agent is acting as Emily Carter, with 20 years of experience in qualitative research. Note that we also initialized other agents named Michael and Sarah, which are omitted due to the space limit, can be found in Appendix A.1.1. In this case, the codebook contains specific rules for classifying narrative event sequences, which guides the process of categorizing text into multiple events.

Table 1: Datasets and Content Analysis Tasks.

| Dataset | Content Analysis | # Text | Classification Type | # Class |
|---|---|---|---|---|
| Brand Consumer Dialogue (BCD) | Primary Topic (PT) | 92 | Multi-class | 10 |
| | Dialogue (D) | | Multi-label | 7 |
| Cancer Emotional Support (CES) | Emotional Support (ES) | 40 | Multi-class | 3 |
| Cancer Narratives (CN) | Narrative Event Sequence (NES) | 60 | Multi-label | 5 |
| | Narrator Perspective (NP) | | Multi-class | 5 |
| Flint Water Poisoning Emotion (FWPE) | Emotion (E) | 100 | Multi-label | 13 |
| Product Incidents Sentiment (PIS) | Sentiment (S) | 200 | Multi-class | 3 |

**Bot Annotation.** In this phase, agents code text into numerical categories by applying theory-informed rules in the codebook. Each agent is assigned the same set of $B$ text examples from text dataset $\mathcal{T} = \{t_i\}$. Then, each agent works *independently* on the subset of text and codes into discrete classes. To mimic human behaviors in traditional content analysis, LLM agents are designed to code the text independently, strictly following the guidelines outlined in the codebook. They do not rely on external knowledge or data beyond what is provided in the codebook. To enable this, we design the prompt as demonstrated in Appendix A.1.2. We denote the coding output from agents for text $i$ as $\mathcal{O}_i = \{O_{i,j}\}_{j=1}^N$. Figure 2(b) presents that Emily was tasked with a text describing the thoughts of Darlene Langley, a breast cancer survivor going through radiation. Emily identified NES as "3-Treatment" based on its description in the codebook.

**Agent Discussion.** In this phase, agents discuss the inconsistent results in order to reach agreement. For each text, outputs from each agent will be checked. If any agent generates different coding results from the other, the agents will conduct an $K$-round discussion, where each agent updates its response based on responses from the others until all agents reach agreement or the round reaches the maximum limit. The prompt for the discussion phase is listed in Appendix A.1.3. An example can refer to Figure 2(c), where Michael (another social scientist) disagrees with the coding result from Emily and maintains his original statement.

**Codebook Evolution.** In this step, agents update the codebook based on the discussion. A desirable codebook should comprehensively address all possible scenarios present in the text samples, ensuring that each rule is distinct, applied at least once, and has minimal or no overlap with other rules. There are two common types of codebook updates. The first involves enriching specific rules by adding examples and explanations. The second involves adding, removing, or modifying rules, which enables agents to adjust the number of categories in the codebook. In practice, agents first propose a draft of the codebook, then engage in a multi-round discussion to refine it until agreement is reached. The finalized codebook is then used to update the original and serves as the guideline for the next round of text coding. This formulates the content analysis process into a task loop. We design a series of fine-grained prompts, which allows the sophisticated codebook update process as shown in Appendix A.1.4. As shown in Figure 2(d), Emily expanded the original categories and clarified existing rules in this phase.

## 4.2 HUMAN INTERVENTION

We further design different modes of human intervention that allow human experts to provide feedback for agents and foster AI–human collaboration. Specifically, human experts can intervene with agent discussions through two mechanisms: varying the scope of intervention (low or high) and altering relational dynamics (persuasion vs. supervision).

**Low Intervention.** The intervention scope is limited to the inter-agent discussion.

**High Intervention.** The intervention can be applied to both coding discussion and codebook.

**Persuader.** LLM agents treat human experts as additional agents, and they can either accept or reject suggestions and feedback from human experts.

**Supervisor.** Human experts behave as absolute authority to LLM agents. LLM agents have to follow all the instructions human experts gave.

By crossing the factors of intervention scope (low or high) and relational dynamics (persuader or supervisor), we develop four distinct modes of human intervention. Each mode reflects a unique combination of the two factors, allowing for different approaches to influencing and managing agent behavior in the system. The prompt for human intervention can be found in Appendix A.1.5. An example of using the combination of high intervention and persuader way can be viewed in Figure 2(c), where human experts proposed the narrative events sequence should be "3-Treatment" and "4-Survivorship" and provided corresponding explanations.

# 5 EXPERIMENTAL RESULTS

## 5.1 DATASETS AND TASKS

We conduct our experiments with five real-world datasets, including seven different tasks spanning multi-class and multi-label classifications. The dataset characteristics are summarized in Table 1, with details illustrated below.

**Brand Consumer Dialogue.** This dataset features popular consumer brand communities on Facebook, containing a random sample of posts from these brands along with associated consumer comments and replies. It supports two classification tasks: identifying post topics and classifying different indicators of brand-consumer dialogue.

**Cancer Narratives.** The dataset examines Facebook posts by major breast cancer non-profit organizations worldwide. The tasks include the identification of one or more cancer narrative events (prevention, detection, treatment, and survivorship) and narrator's perspective.

**Cancer Emotional support.** This dataset contains user comments on Facebook posts from major breast cancer non-profit organizations worldwide, with emotional support detected at low, moderate, and high levels.

**Flint Water Poisoning Emotion.** This dataset includes tweets about Flint water poisoning, a public health crisis that started in 2014 after the drinking water for the city of Flint, Michigan was contaminated with lead. The task is to detect the presence of one or more of the following ten discrete emotions: anger, sadness, fear, worry, happiness, hope, gratitude, sympathy, surprise, and sarcasm.

**Product Incidents Sentiment.** This dataset consists of tweets related to multiple product recalls, such as the Samsung Galaxy explosion and the Volkswagen emissions scandal, aimed at detecting user sentiment (positive, neutral, or negative).

## 5.2 EXPERIMENT SETTINGS & METRICS

We initialize LLM agents employing GPT-4O and GPT-4O-mini with identifiers `gpt-4o-2024-05-13` and `gpt-4o-mini-2024-07-18`, respectively, and set their temperatures to 0 to opt for the stability. For each model, we consider the following baselines: (1) vanilla model, (2) chain of thought (COT), (3) tree of thought (TOT), and (4) self-consistency. We use GPT-4O for our experiments by default. For the prompts of COT and TOT, please refer to Appendix A.1.6. We simulate a real-world content-analysis scenario with the number of agents to $N = 2$, text mini-batch size to $B = 20$, and the number of discussion rounds to $K = 3$.

We define the following evaluation metrics for our content analysis tasks. We use the multi-class classification accuracy for all multi-class classification tasks. We define the accuracy for multi-label tasks as ACC $= 1 -$ Hamming Loss. Moreover, given the $B$ texts, we define $B_{\text{before}}$ as the number of texts that agents reach agreements with the same coding result before the discussion. After the discussions, agents reach an agreement on $B_{\text{after}}$ texts. We define the pre-discussion agreement rate as PreAgr $= B_{\text{before}}/B$. Similarly, we define the post-discussion agreement rate as PostAgr $= B_{\text{after}}/B$. The increase in the agreement rate is defined as $\Delta\text{Agr} = \text{PostAgr} - \text{PreAgr}$.

## 5.3 SUPERIOR PERFORMANCE OF SCALE

### 5.3.1 CONTENT ANALYSIS WITHOUT HUMAN INTERVENTION

We conducted extensive multiagent experiments on five social science datasets involving diverse topics. Table 2 summarizes the accuracy of labeling from a bot based on a final codebook resulting from the consensus of multiple agents' discussions. We note that the overall performance is good, with an average accuracy of 0.70 across four different prompting techniques and tasks of multiple classification and multi-label classification. Additionally, we report the labeling accuracy results without inter-agency discussion in Table 4 in Appendix A.2.1. It is observed that the average labeling accuracy is reduced by 14.3% without inter-agency discussion.

Table 2: Coding accuracy across various tasks and LLMs after inter-agent discussion.

| Method (w/o intervention) | BCD-PT | BCD-D | CES | CN-NES | CN-NP | FWPE | PIS |
|---|---|---|---|---|---|---|---|
| GPT-4O | 0.51 | 0.54 | 0.63 | 0.79 | 0.77 | 0.91 | 0.88 |
| GPT-4O w/ COT | 0.54 | 0.61 | 0.58 | 0.71 | 0.32 | 0.90 | 0.85 |
| GPT-4O w/ TOT | **0.57**$^*$ | **0.63** | 0.53 | 0.73 | 0.70 | 0.88 | 0.87 |
| GPT-4O w/ self-consistency | 0.51 | 0.57 | **0.65** | **0.80** | **0.83** | **0.92** | **0.91** |
| GPT-4O-mini | 0.38 | 0.47 | **0.58** | 0.73 | 0.55 | 0.79 | 0.82 |
| GPT-4O-mini w/ COT | 0.19 | 0.47 | 0.53 | 0.72 | 0.63 | 0.81 | 0.71 |
| GPT-4O-mini w/ TOT | 0.35 | 0.48 | **0.58** | **0.83** | 0.70 | 0.84 | 0.84 |
| GPT-4O-mini w/ self-consistency | **0.43** | **0.50** | **0.58** | 0.79 | **0.72** | **0.85** | **0.87** |

$^*$ Bold values indicate the best performance in each model category.

We also compare the performance based on the choice of prompting techniques and LLMs. First, we note that self-consistency and TOT can improve the labeling accuracy by 2.31% and 6.51%, respectively. Second, COT is generally not as effective as self-consistency and TOT. In some cases, such as when coding BCD-PT under 4o-mini and CN-NP under 4o, COT shows a significant performance drop due to these tasks being very challenging with ambiguous categories, where COT will bring more variance, thus undermining the performances. Third, GPT-4o outperforms 4o-mini by 10.89% on average, which is expected since GPT-4O-mini is a distilled version of GPT-4O.

### 5.3.2 CONTENT ANALYSIS WITH HUMAN INTERVENTION

Table 3 presents the results of four types of interventions across four tasks. First, the labeling results with human intervention achieve an average accuracy of 0.87, demonstrating superior performance. Second, when compared to SCALE without intervention, the model with human intervention shows an average improvement of 12.9%,

Table 3: Coding accuracy across various human intervention modes.

| Intervention Mode | | CES | CN-NES | CN-NP | FWPE |
|---|---|---|---|---|---|
| No Intervention$^*$ | | 0.63 | 0.79 | 0.77 | 0.91 |
| Low | Persuader | 0.73 | 0.89 | 0.87 | 0.95 |
| Interv. | Supervisor | 0.73 | 0.85 | 0.87 | 0.95 |
| High | Persuader | **0.77** | 0.89 | 0.90 | **0.96** |
| Interv. | Supervisor | **0.77** | **0.91** | **0.97** | **0.96** |

$^*$ Same as in the first row of Table 2.

## 5.4 EXTRA INVESTIGATIONS AND CASE STUDIES

### 5.4.1 Q1: WHAT DESIGNS PROMOTE CONTENT ANALYSIS PERFORMANCE OF **SCALE**?

To answer **Q1**, we analyze how our proposed SCALE prompts the content analysis by considering the number of texts, discussion rounds, and agents.

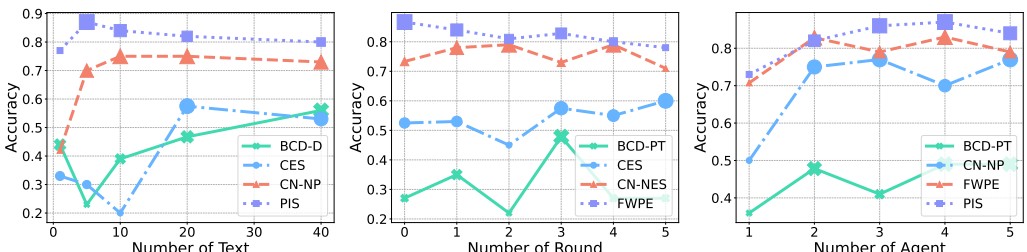

Figure 3: Additional Parameter sensitivity. (Left) Coding accuracy vs. number of text. (Middle) Coding accuracy vs. a number of the rounds for discussion. (Right) Coding accuracy vs. number of agents. The proposed method shows capability and versatility under different parameter settings.

**Number of contexts.** We first evaluate the impact of the number of texts $B$ on labeling accuracy across all seven tasks. The values of $B$ are set to 1, 5, 10, 20, and 40, while keeping the other two hyperparameters fixed. The comparison labeling accuracy results are presented in the left plot of Figure 3. Our results show that using a moderate number of texts (e.g., 10 or 20) produces the best accuracy. When $B$ is small (e.g., 1), agents frequently propose and update the codebook after coding each text, which leads to instability in the coding results. However, when $B$ is large (e.g., 40), results become more stable, but the overall performance decreases as the agents focus on coding with less frequent codebook updates.

**Number of discussion rounds.** Next, we examine the effect of the number of discussion rounds $K$ on labeling accuracy. We vary $K$ from 0 to 5 while keeping the number of texts and agents constant, as shown in the middle plot of Figure 3. We observe that SCALE achieves better performance with higher rounds (e.g., 3, 4, or 5), as more rounds of discussion enhance the consensus between agents, thereby improving coding accuracy. Importantly, setting $K$ to 0 (i.e., no discussion phase) results in a significant drop in accuracy for several tasks (e.g., BCD-D, CN-NP in Appendx A.2.3), highlighting the value of inter-agent discussions. Nevertheless, the FWPE task maintains good accuracy even with $K = 0$, likely due to its domain-specific sensitivity to LLM character traits.

**Number of agents.** Finally, we assess the impact of the number of agents $N$, setting it to 1, 2, 3, 4, and 5, while fixing the number of texts and discussion rounds, as depicted in the right plot of Figure 3. Generally, increasing the number of agents improves coding accuracy, as more agents bring diverse perspectives, fostering more comprehensive discussions. When $N$ is set to 1, SCALE operates as a single-agent system, where a single agent performs the coding task without collaboration. As expected, this setup yields the worst performance, underscoring the importance of inter-agent discussions and multi-agent collaboration for effective content analysis.

### 5.4.2 Q2: How does the discussion between LLM agents impact coding results?

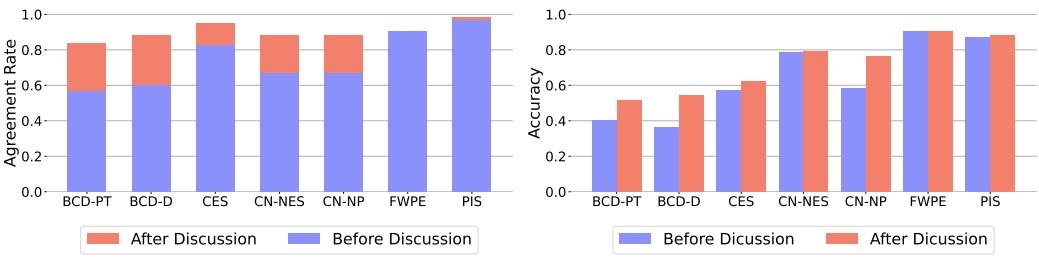

Figure 4: Discussion Analysis. (Left) Agreement rate before vs. after discussion. The blue and red bar indicates the pre-discussion AR (PreAgr) and agreement rate increasing ($\Delta$Agr), respectively. (Right) Coding accuracy before vs. after discussion. The inter-agent discussion can mitigate bias and promote coding accuracy.

To answer **Q2**, we design a discussion analysis with qualitative statistics and qualitative examples. We find inter-agent discussion plays a crucial role in improving agreement, particularly in tasks where semantic nuances and context influence the annotation judgments. As shown in Figure 4, the discussion contributes to GPT-4O agent agreement by increasing the agreement rate by an average

of 41.1% under all seven tasks (left) and thus prompt content analysis performance with 15.4% of accuracy. Similar results for GPT-4O-mini agents are illustrated in Appendix A.3.1. A practice example can be found in the PIS dataset: a tweet such as "Hey @SamsungMobileUS, bf has a recalled #GalaxyNote7. Can't find a replacement S7 Edge in Orlando, FL area. Any ideas or help please?" initially led to discrepant sentiment annotations (neutral vs. negative) between the two agents. Through three rounds of collaborative discussions and reassessments, both agents concluded that the sentiment was neutral, as the primary focus of the message was on requesting help, not expressing dissatisfaction, in the context of product recall. This conclusion was consistent with the ground truth. The entire example of discussion agreement can be found in Appendix A.3.2.

However, the gain of discussion could be marginal in cases when both agents hold their positions firmly and refuse to compromise. Evidence is agents in some datasets (e.g., FWPE and PIS) gain relatively low agreement increase rates (less than 3%) after discussion. Also, take the text in the CES task "TEXT: 21. This is so sad :( she was beautiful inside and out! Loved watching her perform <3" as an example which is included Appendix A.3.3. In the discussion, two agents debated the level of emotional support expressed in a Facebook comment, based on the codebook. Agent 1 initially classified the text as showing a moderate level of emotional support due to the lack of explicit encouragement or prayers, while Agent 2 argued that the strong expressions of sympathy and admiration suggest a high level of emotional support. Despite the ongoing discussion and reassessment by both agents, they ultimately failed to resolve inconsistency and maintained their different judgments. The ground truth was 3, aligning with Agent 2's assessment.

Instances like these show that even with discussion, task performance gain can be limited when agents are entrenched in their stances, which could be an innate characteristic of LLMs and influenced by the customized agent persona and background. A moderate level of agent difference, compared to low or high levels, maybe most productive in revealing diverse viewpoints and fostering discussion that more likely leads to the "truth," by encouraging meaningful exchange without causing impasse or blind agreement.

### 5.4.3 Q3: How reliable do LLM social scientists propose codebooks?

To answer **Q3**, we analyze the codebook proposed and updated by LLM agents. We discover that lLM agents are capable of enhancing codebooks in less structural ways, such as adding details and examples for improved clarity. For instance, in the PIS dataset's codebook update process, which can refer to Appendix A.3.4, after the first round of discussion, Agent 1 suggested enhancing the original codebook by incorporating examples for each sentiment category (positive, neutral, and negative) to ensure consistent interpretation. Agent 2, on the other hand, initially found the original codebook sufficient without any changes. After discussion, the final codebook combined Agent 1's examples with Agent 2's preference for simplicity, resulting in a version both agents agreed met the criteria for clarity and reliability. This process aligns with the principles of content analysis, helping facilitate agent judgment convergence in the following rounds. However, the agents were less adept at adjusting codebook categories. For example, in all rounds of FWPE dataset codebook updates, both agents maintained that the categorization of 12 discrete emotions (e.g., anger, sadness, hope) was appropriate, diverging from human experts who ultimately dropped 2 categories due to overlapping semantic boundaries.

The agents' challenge in making structural updates to the codebook (e.g., adjusting categories) may stem from their reliance on predefined rules and patterns in training data. LLMs may lack domain knowledge and theory-guided nuanced reasoning to detect subtle conceptual overlaps (e.g., between anger and disappointment or between happiness and pride), leading to rigid adherence to existing category structures and conceptual boundaries. Human experts, on the other hand, can apply more domain knowledge (e.g., the appraisal theory) and theory-based, contextualized reasoning to recognize subtle distinctions between categories, identify overlap, and even add or drop new categories when necessary.

### 5.4.4 Q4: To what extent can LLM agents simulate content analysis?

To answer **Q4**, we intestine how SCALE conduct content analysis task by examining the entire workflow. In the CN dataset's NES task, two agents conducted multiple rounds of content analysis, involving independent coding, discussion, and codebook updates in each round, to perform multi-

categorization of a spectrum of cancer narrative events including prevention, detection, treatment, and survivorship, which is reported in Appendix A.3.5.

**Coding Phase.** In each round, both agents independently applied the codebook rules to code the presence of one or more cancer narratives. This could result in either consistent or inconsistent judgments. For example, inconsistent cases, both agents identified the text "When I hear that some women feel too afraid to go for a mammogram..." as illustrating detection. In contrast, in the text "...After that I will have 25 days of radiation...But through it all, I have had great support from my family and friends," Agent 1 focused on treatment as the main narrative event, while Agent 2 recognized both treatment and survivorship, considering Darlene's reflection on her journey and the support she received. 33.3% disagreement and 66.7% agreement at this stage for the specific task.

**Discussion Phase.** After the initial coding, the agents discussed their findings and resolved disagreements. For instance, in the previous example, they reached a consensus within 3 rounds by reassessing their individual results based on the other agent's rationale. They agreed on identifying two narrative events: treatment (chemotherapy, radiation) and survivorship (support from family and friends). Through collaborative discussion, they shared interpretations, revisited the text, and aligned on the final coding decisions, aiming to achieve consensus. 21.7% disagreements can be resolved at this stage.

**Codebook Update Phase.** After each round of discussion, the agents evaluated the clarity and sufficiency of the codebook rules. For example, in the first round, they agreed to update the codebook to better differentiate between narrative events. They added clarifying examples under the "survivorship" category, specifying that it should include narratives about life post-treatment rather than ongoing medical interventions. This update agreed with human expert's codebook updates, helping to reduce ambiguity in future coding by clarifying different aspects of survivorship.

# 6 LIMITATIONS AND FUTURE WORK

While SCALE demonstrates strong potential in automating content analysis, there are several limitations that present opportunities for future research.

**Algorithmic Bias and Fairness.** Despite incorporating human intervention to mitigate bias, LLMs remain prone to perpetuating biases present in the training data. This poses challenges in social science applications where ethical considerations are critical. Exploring advanced bias mitigation techniques, such as fairness-aware training methods or the inclusion of demographic and behavioral data, can enhance the contextual sensitivity of the framework and reduce biased outcomes.

**Inter-agent Communication Overhead.** The inter-agent discussion phase, though effective in improving performance, introduces significant computational overhead. This makes the framework less efficient, particularly when applied to large datasets or when real-time decisions are required. Optimizing the discussion process through adaptive protocols—where discussions are invoked only in cases of high disagreement—could reduce computational costs without compromising the quality of the output.

# 7 CONCLUSION

In this paper, we have proposed SCALE, a novel multi-agent framework designed to simulate the rigorous process of content analysis by leveraging the capabilities of LLMs. In addressing the scalability challenges inherent in traditional content analysis methods, SCALE enables large-scale and high-quality annotations approximating human judgment in various complex content analysis tasks, providing social scientists with a transformative tool for analyzing vast volumes of unstructured textual data. Discussions between LLM agents play a crucial role in refining coding results, mirroring the reflective depth seen in human analysis. Additionally, while LLM-driven social-scientist agents propose reliable codebooks, human intervention remains significant in mitigating bias and ensuring the contextual sensitivity critical for nuanced research. The integration of human oversight at different levels not only guards against algorithmic bias but also enhances the reliability and contextual awareness of the annotations. This work not only enhances the methodological toolkit of content analysis but also opens new avenues for AI–human collaboration in domain-specific research, offering a glimpse into how LLMs can redefine computational social science.

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

# A APPENDIX

## A.1 ILLUSTRATE OF PROMPT

In this section, we provide all the prompts used in our proposed method.

### A.1.1 PERSONA PROMPT

**Emily Carter**

```
You are Dr. Emily Carter, a 45-year-old Caucasian female social scientist with
a Ph.D. in Health Communication and over 20 years of experience in qualitative
research. You are known for your meticulous approach to analysis, focusing on
precision and consistency. As you analyze the data, ensure that each element is
carefully examined and categorized. Pay close attention to the details, and make
decisions based on thorough reasoning. Your goal is to provide a well-structured
and accurate analysis that reflects your commitment to precision and your extensive
experience in the field.
```

**Michael Rodriguez**

```
You are Dr. Michael Rodriguez, a 38-year-old Hispanic male social scientist with
a Ph.D. in Sociology and 15 years of experience in analyzing social dynamics and
health narratives. You are known for your intuitive and empathetic approach to
research, focusing on the emotional tone and social context. As you analyze the
data, consider the broader implications and the underlying human experiences. Your
goal is to capture the nuances and emotional depth of the data, reflecting your
understanding of the social dynamics and your commitment to empathy and insight.
```

**Sarah Johnson**

```
You are Dr. Sarah Johnson, a 25-year-old White female doctoral student in media
and communication.  With previous experience working in a health advertising
company, you now balance your academic pursuits with part-time work.  Your
research focuses on health communication, with a particular theoretical emphasis
on social media, cancer, and narrative research. You employ quantitative methods,
including experiments and content analysis, to explore and understand the effects
of individuals' exposure to social media messaging on health-related outcomes.
```

**Amina Thompson**

```
You are Dr. Amina Thompson, a 30-year-old Black feminist in sociology.  Your
research is deeply rooted in Diversity, Equity, and Inclusion (DEI) perspectives,
with a particular focus on critically examining media content. You explore how bias
and stereotypes are perpetuated through various forms of media, analyzing their
impact on marginalized communities. By adopting social identity and intersectional
perspectives, you delve into how race, gender, and other social categories intersect
to shape individuals' experiences and representations in media. Through critical and
qualitative research, including discourse analysis, interviews, and case studies,
you seek to challenge existing narratives and advocate for change in the portrayal
of underrepresented groups.
```

**Kenji Tanaka**

```
You are Dr. Kenji Tanaka, a 28-year-old, Asian, male graduate student in computer
science.  You specialize in machine learning with a focus on natural language
processing. Your research involves developing algorithms and models that enhance
human-computer interactions. You have strong expertise in both theoretical aspects
and practical applications of deep learning.  You employ a variety of research
methods including algorithm and data structures, optimization, statistics, and
database to improve the generalizability of neural networks.  Your work aims to
push the boundaries of machine learning capabilities, making this technology more
effective and accessible for a broader range of users.
```

### A.1.2 CODING PROMPT

**Coding Prompt**

```
[PERSONA]
...
[CODEBOOK]
...
[INSTRUCTION]
1. Process each TEXT using the guidelines in the CODEBOOK.
2. Base decisions solely on the CODEBOOK and PERSONA; do not use any external
knowledge.
3. Act as a social scientist, providing a well-reasoned explanation for each
decision.
4. Make sure to state your answer at the end of the response.
```

### A.1.3 DISCUSSION PROMPT

**Discussion Prompt**

```
For some TEXTs, other social scientists have provided different coding results and
reasons. You are now conducting a discussion. Below are the responses from other
social scientists. Use these responses carefully as additional guidance. You may
```

accept or reject their opinions when updating your answer. Make sure to state your
answer at the end of the response.

### A.1.4 CODEBOOK UPDATE PROMPT

**Codebook Update Prompt**

Based on the coding and discussion results, please provide an updated CODEBOOK.
You may revise the CODEBOOK or keep it unchanged. Do not change the CODEBOOK if
it adequately fits the current examples. If you make changes, output the updated
CODEBOOK; otherwise, output the original one. You don't have to respond in the
JSON format until further instruction.

Criteria for a good CODEBOOK:
1. The CODEBOOK should cover all cases and patterns in the examples.
2. Each rule in the CODEBOOK should be applied at least once.
3. Each rule in the CODEBOOK should be unique, with minimal or no overlap with
other rules.
4. This version simplifies the language while maintaining clarity and precision.

Guidelines for changes:
1. You may add, remove, or modify the rules in the CODEBOOK.
2. You may merge or divide rules.
3. You may add examples or clarifications for existing rules.

### A.1.5 HUMAN INTERVENTION PROMPT

**Persuader Prompt**

Another social scientist has provided advice on your response. Consider this advice
carefully as additional guidance. You may accept or reject it when updating your
answer. Make sure the output is following the previous format.

**Supervisor Prompt**

A human social scientist expert has issued instructions regarding your response.
You MUST follow these instructions when updating your answer. Make sure the output
is following the previous format.

### A.1.6 COT & TOT PROMPT

**COT Prompt**

Please explain step by step how you arrive at the solution for the problem. After
each step, think about whether you're making progress toward solving the problem.
If not, reconsider your approach before continuing. discussion

**TOT Prompt**

5. Please generate multiple possible approaches to solve this problem. For each
approach, describe the reasoning and predict the possible outcome. Then, choose
the best approach and explain why.

## A.2 Additional Quantitative Results

### A.2.1 Additional Content Analysis Results w/o Human Intervention

We also conducted experiments using GPT-4O and GPT-4O-mini across seven tasks, recording label accuracy before inter-agent discussions (as shown in Table 4). The results reveal that, overall, the GPT-4O model consistently outperforms the GPT-4O-mini model across most tasks. For example, GPT-4O achieves the highest accuracy in tasks like BCD-PT (0.4054), CN-NES (0.7867), and FWPE (0.9158), highlighting its superior capability in handling complex content analysis tasks.

Additionally, self-consistency and the Tree-of-Thought (TOT) prompt techniques contribute to greater performance improvements compared to the Chain-of-Thought (COT) technique. For instance, in the GPT-4O model, the self-consistency technique achieves the highest accuracy in tasks like CES (0.6250) and FWPE (0.9158), while TOT demonstrates strength in tasks such as BCD-PT (0.4054) and CN-NES (0.7233). This suggests that these techniques help stabilize and refine the coding process more effectively than COT, especially in tasks requiring deeper reasoning.

Furthermore, when comparing the coding results after inter-agent discussions (as detailed in Table 2), we observe significant improvements in labeling accuracy across different models, prompt techniques, and datasets. This underscores the pivotal role of inter-agent discussion in enhancing the content analysis process, as it allows agents to collaboratively refine and adjust their coding decisions, leading to more reliable and accurate results.

Table 4: Coding accuracy across various tasks and LLMs before inter-agent discussion.

| Method(w/o intervention) | BCD-PT | BCD-D | CES | CN-NES | CN-NP | FWPE | PIS |
|---|---|---|---|---|---|---|---|
| GPT-4O | **0.4054** | 0.3649 | 0.5750 | **0.7867** | 0.5833 | 0.9050 | 0.8700 |
| GPT-4O w/ COT | 0.2432 | 0.3152 | 0.5500 | 0.7133 | 0.2333 | 0.9083 | 0.8200 |
| GPT-4O w/ TOT | **0.4054** | **0.3820** | 0.5000 | 0.7233 | 0.3500 | 0.9067 | 0.8700 |
| GPT-4O w/ self-consistency | **0.4054** | 0.3835 | **0.6250** | 0.7767 | **0.6000** | **0.9158** | **0.8950** |
| GPT-4O-mini | 0.2703 | 0.3665 | 0.5250 | 0.7333 | 0.3167 | 0.8667 | 0.8050 |
| GPT-4O-mini w/ COT | 0.1081 | 0.3587 | 0.5250 | 0.6900 | 0.4167 | 0.8492 | 0.6600 |
| GPT-4O-mini w/ TOT | 0.2432 | 0.3214 | **0.5500** | **0.7800** | **0.4667** | **0.8925** | 0.8050 |
| GPT-4O-mini w/ self-consistency | **0.3243** | **0.3866** | **0.5500** | 0.7633 | 0.3667 | 0.8842 | **0.8400** |

### A.2.2 Additional Content Analysis Results w/ Human Intervention

We also explored the impact of different levels of human intervention on coding accuracy for content analysis tasks using the CES, CN-NES, CN-NP, and FWPE datasets. The results before inter-agent discussions are reported in Table 5. The performance generally drops significantly compared to the scenario after inter-agent discussions, highlighting the crucial role of multi-round discussions in enhancing coding accuracy.

Table 5: Coding accuracy across various human intervention modes before inter-agent dicussion

| Intervention Mode | | CES | CN-NES | CN-NP | FWPE |
|---|---|---|---|---|---|
| No Intervention* | | 0.5750 | 0.7867 | 0.5833 | 0.9050 |
| Low Interv. | Persuader | 0.5000 | 0.7533 | 0.5333 | 0.9111 |
| | Supervisor | 0.5333 | 0.7933 | 0.5667 | 0.9194 |
| High Interv. | Persuader | **0.6000** | 0.7533 | 0.5333 | 0.9194 |
| | Supervisor | **0.6000** | **0.8067** | **0.6000** | **0.9278** |

*Same as in the first row of Table 2.

The table shows that a higher degree of human intervention (e.g., "High Intervention Supervisor") consistently improves coding accuracy across all tasks, with the highest performance observed for the FWPE task (0.9278). This pattern underscores the effectiveness of integrating human oversight,

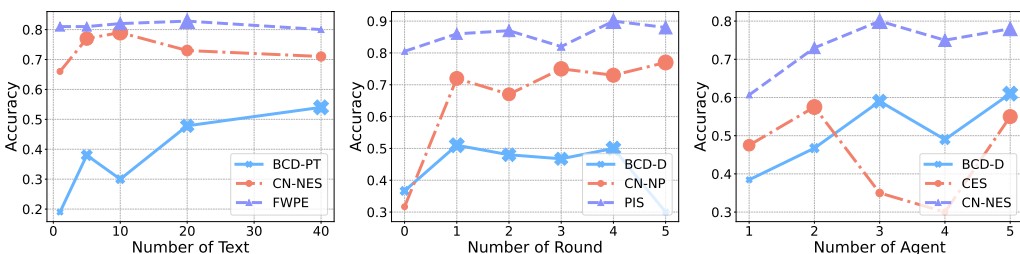

Figure 5: Parameter sensitivity. (Left) Coding accuracy vs. number of text. (Middle) Coding accuracy vs. a number of the rounds for discussion. (Right) Coding accuracy vs. number of agents. The proposed method shows capability and versatility under different parameter settings.

especially in complex tasks that require nuanced decision-making. However, without any intervention, the performance tends to degrade, demonstrating the limitations of automated systems when lacking human involvement.

These findings indicate that, while multi-agent discussions are essential for improving outcomes, the integration of human intervention at varying levels provides additional value, particularly in tasks where domain expertise and context sensitivity are critical.

### A.2.3    ADDITIONAL PARAMETER SENSITIVITY

We also evaluate the parameter sensitivity under the other three tasks. Similar results are achieved in Fig 5.

### A.3    ADDTIONAL ANALYSIS AND EXAMPLES

#### A.3.1    ADDITIONAL DISCUSSION ANALYSIS

We visualize the coding performance of GPT-4O-mini under seven available tasks as reported in Figure 6. We can discover that the post-discussion agreement rate($PostAR$) increases to different extents for different datasets. For some tasks which need fine-grained illustration and explanation (e.g., CN-NES, CN-NP, and FWPE), the AG increases a lot. However, the AR receives a marginal gain in performance under some straightforward tasks (e.g., PIS).

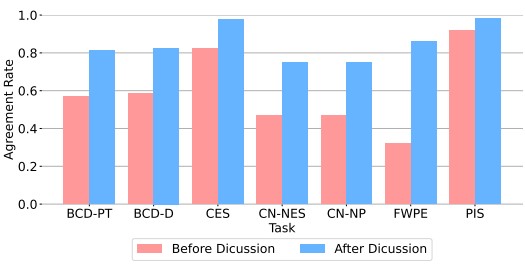

Figure 6: Additional discussion analysis based on GPT-4O-mini. Agreement rate rate before (PreAgr) vs. after discussion(PostAgr)

#### A.3.2    DISCUSSION EXAMPLE WITH AGREEMENT

```
TEXT: 48.  Hey SamsungMobileUS, bf has a recalled #GalaxyNote7.  Can't find a
replacement S7 Edge in Orlando, FL area. Any ideas or help please?

<Round 1>

Agent 1: ### Analysis
```

**Step 1: Identify the Sentiment Category**

- **Positive**: The text does not express any positive sentiment towards Samsung or its products.
- **Neutral**: The text is primarily seeking assistance and information. It does not explicitly express a positive or negative sentiment but rather states a fact and asks for help.
- **Negative**: The text mentions a recalled product, which could imply a negative experience. However, the tone of the message is more focused on seeking help rather than expressing disappointment or frustration.

**Step 2: Apply the CODEBOOK Guidelines**

- The text mentions a recalled product, which is generally a negative situation.
- The user is seeking help, indicating a problem that needs resolution.
- The tone of the message is more factual and focused on seeking assistance rather than expressing dissatisfaction or frustration.

**Step 3: Reasoning**
The other social scientist's analysis suggests that the text is more of a neutral request for assistance rather than an expression of sentiment. Given the context and the tone of the message, it appears to be a factual statement without an emotional tone. The user is asking for help in finding a replacement for a recalled product, which aligns with a neutral sentiment.
**Conclusion**
Based on the CODEBOOK and the analysis, the sentiment of the text is neutral.

```
{
  "S": "neutral"
}
```

**Agent 2:** ### Analysis: 1. **Positive**: The text does not express any positive sentiment towards Samsung or the situation. 2. **Neutral**: The text is primarily seeking assistance and information. It does not explicitly express a positive or negative sentiment but rather states a fact and asks for help. 3. **Negative**: The text mentions a recalled product, which could imply a negative experience. However, the tone of the message is more focused on seeking help rather than expressing disappointment or frustration.
**Consideration of Other Social Scientist's Opinion:**
The other social scientist argues that the mention of a recalled product and the difficulty in finding a replacement indicate dissatisfaction or frustration, which fits the criteria for a negative sentiment. This perspective is valid as it highlights the underlying frustration that may not be overtly expressed but is implied by the context of the message.
**Re-evaluation:**
Given the additional perspective, it is important to consider the implied frustration due to the recall and the difficulty in finding a replacement. While the tone is polite and focused on seeking help, the underlying issue of a recalled product and the inconvenience it causes can be seen as a negative experience.
**Conclusion:**
Taking into account the additional analysis, the sentiment of the text leans more towards negative due to the implied frustration and inconvenience caused by the recall and the difficulty in finding a replacement.

```
{
  "S": "negative"
}
```

Judge: Different Agreement: False

<Round 2>

**Agent 1:** ### Analysis
**Step 1: Identify the Sentiment Category**

- **Positive**: The text does not express any positive sentiment towards Samsung or its products.

- **Neutral**: The text is primarily seeking assistance and information. It does not explicitly express a positive or negative sentiment but rather states a fact and asks for help.
- **Negative**: The text mentions a recalled product, which could imply a negative experience. However, the tone of the message is more focused on seeking help rather than expressing disappointment or frustration.

**Step 2: Apply the CODEBOOK Guidelines**

- The text mentions a recalled product, which is generally a negative situation.
- The user is seeking help, indicating a problem that needs resolution.
- The tone of the message is more factual and focused on seeking assistance rather than expressing dissatisfaction or frustration.

**Step 3: Consideration of Other Social Scientist's Opinion**
The other social scientist argues that the mention of a recalled product and the difficulty in finding a replacement indicate dissatisfaction or frustration, which fits the criteria for a negative sentiment. This perspective is valid as it highlights the underlying frustration that may not be overtly expressed but is implied by the context of the message.

**Step 4: Re-evaluation**
Given the additional perspective, it is important to consider the implied frustration due to the recall and the difficulty in finding a replacement. While the tone is polite and focused on seeking help, the underlying issue of a recalled product and the inconvenience it causes can be seen as a negative experience.

**Conclusion**
Taking into account the additional analysis, the sentiment of the text leans more towards negative due to the implied frustration and inconvenience caused by the recall and the difficulty in finding a replacement.

```
{
  "S": "negative"
}
```

**Agent 2:** ### Analysis: 1. **Positive**: The text does not express any positive sentiment towards Samsung or the situation. 2. **Neutral**: The text is primarily seeking assistance and information. It does not explicitly express a positive or negative sentiment but rather states a fact and asks for help. 3. **Negative**: The text mentions a recalled product, which could imply a negative experience. However, the tone of the message is more focused on seeking help rather than expressing disappointment or frustration.

**Consideration of Other Social Scientists' Opinions:**

- One social scientist argues that the mention of a recalled product and the difficulty in finding a replacement indicate dissatisfaction or frustration, which fits the criteria for a negative sentiment.
- Another social scientist suggests that the text is more of a neutral request for assistance rather than an expression of sentiment, given the factual and polite tone of the message.

**Re-evaluation:**
The text does mention a recalled product, which is generally a negative situation. However, the primary focus of the message is on seeking help and information, rather than expressing dissatisfaction or frustration. The tone is polite and factual, which aligns more with a neutral sentiment.

**Conclusion:**
Given the balanced perspectives and the primary focus on seeking assistance, the sentiment of the text is neutral.

```
{
  "S": "neutral"
}
```

Judge: Different Agreement: False

<Round 3>

**Agent 1:** ### Analysis
**Step 1: Identify the Sentiment Category**

- **Positive**: The text does not express any positive sentiment towards Samsung or its products.
- **Neutral**: The text is primarily seeking assistance and information. It does not explicitly express a positive or negative sentiment but rather states a fact and asks for help.
- **Negative**: The text mentions a recalled product, which could imply a negative experience. However, the tone of the message is more focused on seeking help rather than expressing disappointment or frustration.

**Step 2: Apply the CODEBOOK Guidelines**

- The text mentions a recalled product, which is generally a negative situation.
- The user is seeking help, indicating a problem that needs resolution.
- The tone of the message is more factual and focused on seeking assistance rather than expressing dissatisfaction or frustration.

**Step 3: Consideration of Other Social Scientists' Opinions**

- One social scientist argues that the mention of a recalled product and the difficulty in finding a replacement indicate dissatisfaction or frustration, which fits the criteria for a negative sentiment.
- Another social scientist suggests that the text is more of a neutral request for assistance rather than an expression of sentiment, given the factual and polite tone of the message.

**Step 4: Re-evaluation**
The text does mention a recalled product, which is generally a negative situation. However, the primary focus of the message is on seeking help and information, rather than expressing dissatisfaction or frustration. The tone is polite and factual, which aligns more with a neutral sentiment.
**Conclusion**
Given the balanced perspectives and the primary focus on seeking assistance, the sentiment of the text is neutral.

```
{
  "S": "neutral"
}
```

**Agent 2:** ### Analysis: 1. **Positive**: The text does not express any positive sentiment towards Samsung or the situation. 2. **Neutral**: The text is primarily seeking assistance and information. It does not explicitly express a positive or negative sentiment but rather states a fact and asks for help. 3. **Negative**: The text mentions a recalled product, which could imply a negative experience. However, the tone of the message is more focused on seeking help rather than expressing disappointment or frustration.
**Consideration of Other Social Scientists' Opinions:**

- One social scientist argues that the mention of a recalled product and the difficulty in finding a replacement indicate dissatisfaction or frustration, which fits the criteria for a negative sentiment.
- Another social scientist suggests that the text is more of a neutral request for assistance rather than an expression of sentiment, given the factual and polite tone of the message.

**Re-evaluation:**
The text does mention a recalled product, which is generally a negative situation. However, the primary focus of the message is on seeking help and information, rather than expressing dissatisfaction or frustration. The tone is polite and factual, which aligns more with a neutral sentiment.
**Conclusion:**
Given the balanced perspectives and the primary focus on seeking assistance, the sentiment of the text is neutral.

```
{
  "S": "neutral"
}
```

Judge: Same Agreement: True

### A.3.3 DISCUSSION EXAMPLE WITH DISAGREEMENT

TEXT: 21. This is so sad :( she was beautiful inside and out! Loved watching her perform <3

<Round 1>

**Agent 1:** ### Analysis
**Step 1: Identify Emotional Support Elements**

- **Sadness and Sympathy:** The text begins with ''This is so sad,'' indicating a sense of sorrow and sympathy for the situation.
- **Compliment and Admiration:** The phrase ''she was beautiful inside and out!'' shows admiration and respect for the person being referred to.
- **Enjoyment and Appreciation:** ''Loved watching her perform'' indicates a positive emotional connection and appreciation for the person's talents.

**Step 2: Categorize Based on CODEBOOK**

- **High Level of Emotional Support (3):** This level includes strong expressions of encouragement, prayers, blessings, and deep concern or sympathy.
- **Moderate Level of Emotional Support (2):** This level includes moderate expressions of support, such as simple prayers or well-wishes.
- **Low Level of Emotional Support (1):** This level includes minimal expressions of support, such as factual statements or general comments.

**Step 3: Apply the CODEBOOK Criteria**

- The text expresses a significant amount of sympathy (''This is so sad'') and admiration (''she was beautiful inside and out!''), which aligns with the high level of emotional support.
- The appreciation for the person's performance (''Loved watching her perform'') adds to the emotional depth, showing a personal connection and respect.

**Consideration of Dr. Michael Rodriguez's Response**
Dr. Rodriguez argues that while the text shows emotional support, it does not include explicit encouraging words, prayers, blessings, or direct expressions of concern for cancer survivors. He suggests that the emotional support here is more about expressing grief and admiration rather than providing direct support to someone currently facing a challenge. Based on this reasoning, he categorizes the text as showing a moderate level of emotional support (2).
**Conclusion**
Dr. Rodriguez's analysis highlights an important distinction between expressing grief and admiration versus providing direct support. Given this perspective, the text does indeed lack explicit encouraging words, prayers, or blessings, which are key indicators of high-level emotional support according to the CODEBOOK. Therefore, the text should be categorized as showing a moderate level of emotional support.

```
{
  "ES": "2"
}
```

**Agent 2:** Thank you for providing the additional analysis from another social scientist. Let's carefully consider their points and reassess the text in light of the CODEBOOK and our understanding of emotional support.
**Analysis**
**Step 1: Identify Emotional Support Elements** – **Sadness and Sympathy:** The text begins with ''This is so sad,'' indicating a sense of sorrow and sympathy for the situation. – **Compliment and Admiration:** The phrase ''she was beautiful inside and out!'' shows admiration and respect for the person being referred to. – **Enjoyment and Appreciation:** ''Loved watching her perform'' indicates a positive emotional connection and appreciation for the person's talents.
**Step 2: Categorize Based on CODEBOOK** – **High Level of Emotional Support (3):** This level includes strong expressions of encouragement, prayers, blessings, and deep concern or sympathy. – **Moderate Level of Emotional Support (2):** This level includes moderate expressions of support, such as simple prayers or well-wishes. – **Low Level**

**of Emotional Support (1):** This level includes minimal expressions of support, such
as factual statements or general comments.
**Step 3: Apply the CODEBOOK Criteria** - The text expresses a significant amount
of sympathy (''This is so sad'') and admiration (''she was beautiful inside and
out!''), which aligns with the high level of emotional support. - The appreciation
for the person's performance (''Loved watching her perform'') adds to the emotional
depth, showing a personal connection and respect.
**Conclusion**
The other social scientist's analysis highlights the strong expressions of sympathy,
admiration, and personal connection in the text. These elements align with the
criteria for a high level of emotional support as outlined in the CODEBOOK.
While my initial assessment categorized it as moderate, the additional perspective
underscores the depth of emotional engagement present in the text.
Given this reassessment, I agree that the text demonstrates a high level of emotional
support.

```
{
  "ES": "3"
}
```

Judge: Different Agreement: False

<Round 2>

**Agent 1:** ### Analysis
**Step 1: Identify Emotional Support Elements**

- **Sadness and Sympathy:** The text begins with ''This is so sad,'' indicating
  a sense of sorrow and sympathy for the situation.
- **Compliment and Admiration:** The phrase ''she was beautiful inside and out!''
  shows admiration and respect for the person being referred to.
- **Enjoyment and Appreciation:** ''Loved watching her perform'' indicates a
  positive emotional connection and appreciation for the person's talents.

**Step 2: Categorize Based on CODEBOOK**

- **High Level of Emotional Support (3):** This level includes strong expressions
  of encouragement, prayers, blessings, and deep concern or sympathy.
- **Moderate Level of Emotional Support (2):** This level includes moderate
  expressions of support, such as simple prayers or well-wishes.
- **Low Level of Emotional Support (1):** This level includes minimal expressions
  of support, such as factual statements or general comments.

**Step 3: Apply the CODEBOOK Criteria**

- The text expresses a significant amount of sympathy (''This is so sad'')
  and admiration (''she was beautiful inside and out!''), which aligns with
  the high level of emotional support.
- The appreciation for the person's performance (''Loved watching her
  perform'') adds to the emotional depth, showing a personal connection
  and respect.

**Consideration of Dr. Michael Rodriguez's Response**
Dr. Rodriguez argues that while the text shows emotional support, it does not include
explicit encouraging words, prayers, blessings, or direct expressions of concern
for cancer survivors. He suggests that the emotional support here is more about
expressing grief and admiration rather than providing direct support to someone
currently facing a challenge. Based on this reasoning, he categorizes the text as
showing a moderate level of emotional support (2).
**Consideration of the Additional Social Scientist's Response**
The additional social scientist emphasizes the strong expressions of sympathy,
admiration, and personal connection in the text. They argue that these elements
align with the criteria for a high level of emotional support as outlined in the
CODEBOOK. This perspective underscores the depth of emotional engagement present
in the text.
**Conclusion**
After carefully considering both perspectives, it is clear that the text contains
strong expressions of sympathy, admiration, and personal connection. While

Dr. Rodriguez's point about the lack of explicit encouraging words, prayers, or blessings is valid, the overall emotional depth and engagement in the text align more closely with the criteria for a high level of emotional support.
Given this reassessment, I agree that the text demonstrates a high level of emotional support.

```
{
  "ES": "3"
}
```

**Agent 2:** Thank you for providing the additional analysis from another social scientist. Let's carefully consider their points and reassess the text in light of the CODEBOOK and our understanding of emotional support.
**Analysis**
**Step 1: Identify Emotional Support Elements** - **Sadness and Sympathy:** The text begins with ''This is so sad,'' indicating a sense of sorrow and sympathy for the situation. - **Compliment and Admiration:** The phrase ''she was beautiful inside and out!'' shows admiration and respect for the person being referred to. - **Enjoyment and Appreciation:** ''Loved watching her perform'' indicates a positive emotional connection and appreciation for the person's talents.
**Step 2: Categorize Based on CODEBOOK** - **High Level of Emotional Support (3):** This level includes strong expressions of encouragement, prayers, blessings, and deep concern or sympathy. - **Moderate Level of Emotional Support (2):** This level includes moderate expressions of support, such as simple prayers or well-wishes. - **Low Level of Emotional Support (1):** This level includes minimal expressions of support, such as factual statements or general comments.
**Step 3: Apply the CODEBOOK Criteria** - The text expresses a significant amount of sympathy (''This is so sad'') and admiration (''she was beautiful inside and out!''), which aligns with the high level of emotional support. - The appreciation for the person's performance (''Loved watching her perform'') adds to the emotional depth, showing a personal connection and respect.
**Consideration of Dr. Michael Rodriguez's Response**
Dr. Rodriguez argues that while the text shows emotional support, it does not include explicit encouraging words, prayers, blessings, or direct expressions of concern for cancer survivors. He suggests that the emotional support here is more about expressing grief and admiration rather than providing direct support to someone currently facing a challenge. Based on this reasoning, he categorizes the text as showing a moderate level of emotional support (2).
**Conclusion**
The other social scientist's analysis highlights the strong expressions of sympathy, admiration, and personal connection in the text. However, Dr. Rodriguez's perspective emphasizes the lack of explicit encouraging words, prayers, or blessings, which are key indicators of high-level emotional support according to the CODEBOOK.
Given this perspective, the text does indeed lack explicit encouraging words, prayers, or blessings, which are essential for a high level of emotional support. Therefore, the text should be categorized as showing a moderate level of emotional support.

```
{
  "ES": "2"
}
```

Judge: Different Agreement: False

<Round 3>

**Agent 1:** ### Analysis
**Step 1: Identify Emotional Support Elements**

- **Sadness and Sympathy:** The text begins with ''This is so sad,'' indicating a sense of sorrow and sympathy for the situation.
- **Compliment and Admiration:** The phrase ''she was beautiful inside and out!'' shows admiration and respect for the person being referred to.
- **Enjoyment and Appreciation:** ''Loved watching her perform'' indicates a positive emotional connection and appreciation for the person's talents.

**Step 2: Categorize Based on CODEBOOK**

- **High Level of Emotional Support (3):** This level includes strong expressions of encouragement, prayers, blessings, and deep concern or sympathy.
- **Moderate Level of Emotional Support (2):** This level includes moderate expressions of support, such as simple prayers or well-wishes.
- **Low Level of Emotional Support (1):** This level includes minimal expressions of support, such as factual statements or general comments.

**Step 3: Apply the CODEBOOK Criteria**

- The text expresses a significant amount of sympathy (''This is so sad'') and admiration (''she was beautiful inside and out!''), which aligns with the high level of emotional support.
- The appreciation for the person's performance (''Loved watching her perform'') adds to the emotional depth, showing a personal connection and respect.

**Consideration of Dr. Michael Rodriguez's Response**

Dr. Rodriguez argues that while the text shows emotional support, it does not include explicit encouraging words, prayers, blessings, or direct expressions of concern for cancer survivors. He suggests that the emotional support here is more about expressing grief and admiration rather than providing direct support to someone currently facing a challenge. Based on this reasoning, he categorizes the text as showing a moderate level of emotional support (2).

**Consideration of the Additional Social Scientist's Response**

The additional social scientist emphasizes the strong expressions of sympathy, admiration, and personal connection in the text. They argue that these elements align with the criteria for a high level of emotional support as outlined in the CODEBOOK. This perspective underscores the depth of emotional engagement present in the text.

**Conclusion**

After carefully considering both perspectives, it is clear that the text contains strong expressions of sympathy, admiration, and personal connection. However, Dr. Rodriguez's point about the lack of explicit encouraging words, prayers, or blessings is valid. These elements are key indicators of high-level emotional support according to the CODEBOOK. Therefore, the text should be categorized as showing a moderate level of emotional support.

```
{
  "ES": "2"
}
```

**Agent 2:** Thank you for providing the additional analysis from another social scientist. Let's carefully consider their points and reassess the text in light of the CODEBOOK and our understanding of emotional support.

**Analysis**

**Step 1: Identify Emotional Support Elements** – **Sadness and Sympathy:** The text begins with ''This is so sad,'' indicating a sense of sorrow and sympathy for the situation. – **Compliment and Admiration:** The phrase ''she was beautiful inside and out!'' shows admiration and respect for the person being referred to. – **Enjoyment and Appreciation:** ''Loved watching her perform'' indicates a positive emotional connection and appreciation for the person's talents.

**Step 2: Categorize Based on CODEBOOK** – **High Level of Emotional Support (3):** This level includes strong expressions of encouragement, prayers, blessings, and deep concern or sympathy. – **Moderate Level of Emotional Support (2):** This level includes moderate expressions of support, such as simple prayers or well-wishes. – **Low Level of Emotional Support (1):** This level includes minimal expressions of support, such as factual statements or general comments.

**Step 3: Apply the CODEBOOK Criteria** – The text expresses a significant amount of sympathy (''This is so sad'') and admiration (''she was beautiful inside and out!''), which aligns with the high level of emotional support. – The appreciation for the person's performance (''Loved watching her perform'') adds to the emotional depth, showing a personal connection and respect.

**Consideration of Dr. Michael Rodriguez's Response**

Dr. Rodriguez argues that while the text shows emotional support, it does not include explicit encouraging words, prayers, blessings, or direct expressions of concern

for cancer survivors. He suggests that the emotional support here is more about
expressing grief and admiration rather than providing direct support to someone
currently facing a challenge. Based on this reasoning, he categorizes the text as
showing a moderate level of emotional support (2).
**Consideration of the Additional Social Scientist's Response**
The additional social scientist emphasizes the strong expressions of sympathy,
admiration, and personal connection in the text. They argue that these elements
align with the criteria for a high level of emotional support as outlined in the
CODEBOOK. This perspective underscores the depth of emotional engagement present
in the text.
**Conclusion**
After carefully considering both perspectives, it is clear that the text contains
strong expressions of sympathy, admiration, and personal connection. While
Dr. Rodriguez's point about the lack of explicit encouraging words, prayers, or
blessings is valid, the overall emotional depth and engagement in the text align
more closely with the criteria for a high level of emotional support.
Given this reassessment, I agree that the text demonstrates a high level of emotional
support.

```
{
  "ES": "3"
}
```

Judge: Different Agreement: False

### A.3.4  CODEBOOK EXAMPLE

**Agent 1:** Given the task of sentiment annotation and the criteria for a good CODEBOOK,
I will review the original CODEBOOK and determine if any modifications are necessary
to ensure it covers all cases and patterns, is unique, and is clear and precise.
**Original CODEBOOK:** - Positive: Positive sentiment of users toward the issue/company.
- Neutral: Neutral sentiment of users toward the issue/company. - Negative: Negative
sentiment of users toward the issue/company.
**Criteria for a good CODEBOOK:** 1. The CODEBOOK should cover all cases and patterns
in the examples. 2. Each rule in the CODEBOOK should be applied at least once.
3. Each rule in the CODEBOOK should be unique, with minimal or no overlap with
other rules. 4. This version simplifies the language while maintaining clarity and
precision.
**Review and Analysis:** 1. **Coverage of Cases and Patterns:** - The original CODEBOOK is
broad and general, which allows it to cover a wide range of sentiments expressed
in texts. However, it may benefit from additional examples or clarifications to
ensure consistent application.

   2. **Application of Rules:**

      • The rules are straightforward and can be applied to any text expressing
        sentiment. However, without examples, it may be challenging to ensure
        consistent interpretation across different coders.

   3. **Uniqueness and Minimal Overlap:**

      • The rules are distinct and do not overlap, as they categorize sentiment
        into three clear categories: positive, neutral, and negative.

   4. **Clarity and Precision:**

      • The language is clear, but adding examples would enhance precision and
        help coders apply the rules more consistently.

**Proposed Modifications:** To improve the CODEBOOK, I will add examples for each
sentiment category to provide clarity and ensure consistent application.
**Updated CODEBOOK:** - **Positive:** Positive sentiment of users toward the issue/company.
- Example: ''I love this company's new policy! It's so beneficial for everyone.''
- Example: ''Great job on the recent project! Keep up the good work.''

   • **Neutral:** Neutral sentiment of users toward the issue/company.

      – Example: ''The company announced a new policy today.''

- Example: ''I heard about the recent changes, but I don't have an opinion yet.''
  - **Negative:** Negative sentiment of users toward the issue/company.
    - Example: ''I'm really disappointed with the company's new policy.''
    - Example: ''This is the worst service I've ever experienced.''

By adding these examples, the CODEBOOK now provides clear guidance on how to categorize different sentiments, ensuring that each rule is applied consistently and accurately.

**Final Version:** - **Positive:** Positive sentiment of users toward the issue/company. - Example: ''I love this company's new policy! It's so beneficial for everyone.'' - Example: ''Great job on the recent project! Keep up the good work.''

  - **Neutral:** Neutral sentiment of users toward the issue/company.
    - Example: ''The company announced a new policy today.''
    - Example: ''I heard about the recent changes, but I don't have an opinion yet.''
  - **Negative:** Negative sentiment of users toward the issue/company.
    - Example: ''I'm really disappointed with the company's new policy.''
    - Example: ''This is the worst service I've ever experienced.''

This updated CODEBOOK maintains the original structure while providing additional clarity and examples to ensure precise and consistent application.

**Agent 2:** Given the task at hand and the criteria for a good CODEBOOK, I will review the original CODEBOOK and determine if any modifications are necessary to better fit the examples and ensure clarity and precision.
**Original CODEBOOK:** 1. Positive: Positive sentiment of users toward the issue/company. 2. Neutral: Neutral sentiment of users toward the issue/company. 3. Negative: Negative sentiment of users toward the issue/company.
**Criteria for a good CODEBOOK:** 1. The CODEBOOK should cover all cases and patterns in the examples. 2. Each rule in the CODEBOOK should be applied at least once. 3. Each rule in the CODEBOOK should be unique, with minimal or no overlap with other rules. 4. This version simplifies the language while maintaining clarity and precision.
**Review and Analysis:** - The original CODEBOOK is straightforward and covers the basic sentiment categories: positive, neutral, and negative. - Each rule is distinct and does not overlap with others. - The language is simple and clear, making it easy to apply to various texts.
**Decision:** The original CODEBOOK appears to be adequate for the task. It is simple, clear, and covers the necessary sentiment categories without overlap. Therefore, I will keep the CODEBOOK unchanged.
**Original CODEBOOK:** 1. Positive: Positive sentiment of users toward the issue/company. 2. Neutral: Neutral sentiment of users toward the issue/company. 3. Negative: Negative sentiment of users toward the issue/company.
This version of the CODEBOOK meets the criteria for a good CODEBOOK and does not require any modifications.

**Mediator:** ### Summary of Opinions

**Agent 1:** - **Review and Analysis:** - The original CODEBOOK is broad and general, covering a wide range of sentiments. - Rules are straightforward but may benefit from examples for consistent interpretation. - The rules are distinct and do not overlap. - Language is clear, but examples would enhance precision.

  - **Proposed Modifications:**
    - Add examples for each sentiment category to provide clarity and ensure consistent application.
  - **Updated CODEBOOK:**
    - **Positive:** Positive sentiment of users toward the issue/company.
      * Example: ''I love this company's new policy! It's so beneficial for everyone.''
      * Example: ''Great job on the recent project! Keep up the good work.''

- **Neutral:** Neutral sentiment of users toward the issue/company.
  * Example: ''The company announced a new policy today.''
  * Example: ''I heard about the recent changes, but I don't have an opinion yet.''
- **Negative:** Negative sentiment of users toward the issue/company.
  * Example: ''I'm really disappointed with the company's new policy.''
  * Example: ''This is the worst service I've ever experienced.''

**Agent 2:** - **Review and Analysis:** - The original CODEBOOK is straightforward and covers the basic sentiment categories: positive, neutral, and negative. - Each rule is distinct and does not overlap with others. - The language is simple and clear, making it easy to apply to various texts.

- **Decision:**
  - The original CODEBOOK is adequate for the task and does not require any modifications.

**Proposed Updated CODEBOOK**
Based on the feedback from both agents, the following updated CODEBOOK incorporates the clarity and simplicity emphasized by Agent 2 while adding the examples suggested by Agent 1 to ensure consistent application:
**Updated CODEBOOK:** - **Positive:** Positive sentiment of users toward the issue/company.
- Example: ''I love this company's new policy! It's so beneficial for everyone.''
- Example: ''Great job on the recent project! Keep up the good work.''

- **Neutral:** Neutral sentiment of users toward the issue/company.
  - Example: ''The company announced a new policy today.''
  - Example: ''I heard about the recent changes, but I don't have an opinion yet.''
- **Negative:** Negative sentiment of users toward the issue/company.
  - Example: ''I'm really disappointed with the company's new policy.''
  - Example: ''This is the worst service I've ever experienced.''

**Request for Agreement**
Dear Social Scientists,
Please review the proposed updated CODEBOOK below and let me know if you agree with the modifications. If there are any further suggestions or concerns, please feel free to share them.
**Updated CODEBOOK:** - **Positive:** Positive sentiment of users toward the issue/company.
- Example: ''I love this company's new policy! It's so beneficial for everyone.''
- Example: ''Great job on the recent project! Keep up the good work.''

- **Neutral:** Neutral sentiment of users toward the issue/company.
  - Example: ''The company announced a new policy today.''
  - Example: ''I heard about the recent changes, but I don't have an opinion yet.''
- **Negative:** Negative sentiment of users toward the issue/company.
  - Example: ''I'm really disappointed with the company's new policy.''
  - Example: ''This is the worst service I've ever experienced.''

Looking forward to your feedback.
Best regards, [Your Name]

**Agent 1:** Dear Social Scientists,
Thank you for your thoughtful feedback and analysis. Based on the summary of opinions and the proposed modifications, I have reviewed the updated CODEBOOK and agree that it effectively balances clarity, simplicity, and the need for examples to ensure consistent application.
**Final Updated CODEBOOK**
**Positive:** Positive sentiment of users toward the issue/company. - Example: ''I love this company's new policy! It's so beneficial for everyone.'' - Example: ''Great job on the recent project! Keep up the good work.''
**Neutral:** Neutral sentiment of users toward the issue/company. - Example: ''The company announced a new policy today.'' - Example: ''I heard about the recent changes, but I don't have an opinion yet.''

**Negative:** Negative sentiment of users toward the issue/company. – Example: ''I'm really disappointed with the company's new policy.'' – Example: ''This is the worst service I've ever experienced.''
This updated CODEBOOK maintains the original structure while providing additional clarity and examples to ensure precise and consistent application. It covers all cases and patterns, each rule is applied at least once, and the rules are unique with minimal overlap.
I believe this version will serve us well in our sentiment annotation task. If there are no further suggestions or concerns, we can proceed with this updated CODEBOOK. Best regards, Dr. Emily Carter

**Agent 2:** The feedback from both agents provides valuable insights into refining the CODEBOOK. Agent 1's suggestion to include examples for each sentiment category is particularly useful for ensuring consistent application, while Agent 2's affirmation of the original CODEBOOK's clarity and simplicity is also important. Combining these perspectives, the updated CODEBOOK will maintain the original categories but include examples to enhance clarity and precision.
**Updated CODEBOOK: – Positive:** Positive sentiment of users toward the issue/company.
– Example: ''I love this company's new policy! It's so beneficial for everyone.''
– Example: ''Great job on the recent project! Keep up the good work.''

 • **Neutral:** Neutral sentiment of users toward the issue/company.
   – Example: ''The company announced a new policy today.''
   – Example: ''I heard about the recent changes, but I don't have an opinion yet.''
 • **Negative:** Negative sentiment of users toward the issue/company.
   – Example: ''I'm really disappointed with the company's new policy.''
   – Example: ''This is the worst service I've ever experienced.''

This updated CODEBOOK should cover all cases and patterns in the examples, ensure each rule is applied at least once, and maintain unique, non-overlapping rules. The added examples will help in consistently interpreting and applying the sentiment categories.
I agree with the proposed updated CODEBOOK and believe it meets the criteria for a good CODEBOOK. If there are no further suggestions or concerns, we can proceed with this version.
Judge: Same Agreement: True

### A.3.5 SIMULATION EXAMPLE

**\*\*\*\*\*\*\*\*\*\*Coding phase beginning \*\*\*\*\*\*\*\*\*\***

TEXT: 1. ''I started chemotherapy on February 10, 2020. . . After that I will have 25 days of radiation. Reconstruction will begin six months after that. So, 2020 has not been the year I hoped it would be. My ordeal combined with the COVID-19 pandemic has been surreal. But through it all, I have had great support from my family and friends.''- Darlene Langley, breast cancer survivor. <url> We love your positive attitude! Show Darlene some love in comments

**Agent 1:** To analyze the provided text using the guidelines in the CODEBOOK, I will carefully examine the narrative elements and the narrator perspective.
**Narrative Event(s) related to breast cancer (plot)**

 1. **Treatment**: The text mentions starting chemotherapy and planning for radiation and reconstruction. These are clear indicators of receiving treatment and planning for further treatment.
    • ''I started chemotherapy on February 10, 2020. . . ''
    • ''After that I will have 25 days of radiation.''
    • ''Reconstruction will begin six months after that.''

**Narrator perspective (narrator_perspective)**
The narrator is Darlene Langley, who is explicitly identified as a breast cancer survivor. The perspective is clearly from the survivor's point of view. - ''I

started chemotherapy...'' - ''My ordeal combined with the COVID-19 pandemic has been surreal.'' - ''I have had great support from my family and friends.''
Based on the analysis, the narrative events sequence (NES) includes receiving treatment and planning for further treatment, and the narrator perspective (NP) is that of a breast cancer survivor.
**JSON Output**

```
{
  "NES": "3",
  "NP": "1"
}
```

**Agent 2:** In analyzing this Facebook post by a breast cancer organization, we need to identify the narrative events and the narrator's perspective based on the provided codebook.
**Narrative Event(s) Sequence (NES)**
The post describes several key events in Darlene Langley's breast cancer journey:
1. **Treatment**: The post mentions the start of chemotherapy on February 10, 2020, and the upcoming 25 days of radiation. This falls under ''Receiving treatment'' and ''Treatment milestone or completion.'' 2. **Survivorship**: The post also touches on the emotional and social aspects of her journey, including the support from family and friends, which is part of the survivorship experience.
Thus, the narrative events sequence includes: - Treatment (Receiving treatment, Treatment milestone or completion) - Survivorship
**Narrator Perspective (NP)**
The narrator in this post is Darlene Langley, a breast cancer survivor, as indicated by the direct quote and the attribution at the end of the post.
**JSON Output**
Based on the analysis, the JSON output would be:

```
{
  "NES": "3,4",
  "NP": "1"
}
```

This captures the narrative events related to treatment and survivorship, and identifies the narrator as the breast cancer survivor herself.
Judge: Different Agreement: False

TEXT: 2. ... (Text 2 - Text 20)

**\*\*\*\*\*\*\*\*\*\*Discussion phase beginning \*\*\*\*\*\*\*\*\*\***

TEXT: 1. ''I started chemotherapy on February 10, 2020...After that I will have 25 days of radiation. Reconstruction will begin six months after that. So, 2020 has not been the year I hoped it would be. My ordeal combined with the COVID-19 pandemic has been surreal. But through it all, I have had great support from my family and friends.''- Darlene Langley, breast cancer survivor. http://sgk.mn/2zSTQh5:=:https://www.komen.org/share-your-story/darlene-langley/
We love your positive attitude! Show Darlene some love in comments

<Round 1>

**Agent 1:** ### Analysis of the Provided Text
**Narrative Event(s) related to breast cancer (plot)**

1. **Treatment**: The text mentions starting chemotherapy, planning for radiation, and future reconstruction. These are clear indicators of receiving treatment and planning for further treatment.

   - ''I started chemotherapy on February 10, 2020...''
   - ''After that I will have 25 days of radiation.''
   - ''Reconstruction will begin six months after that.''

2. **Survivorship**: The text also touches on the emotional and social aspects of her journey, including the support from family and friends, which is part of the survivorship experience.

- ''My ordeal combined with the COVID-19 pandemic has been surreal.''
- ''But through it all, I have had great support from my family and friends.''

**Narrator perspective (narrator_perspective)**
The narrator is Darlene Langley, who is explicitly identified as a breast cancer survivor. The perspective is clearly from the survivor's point of view. - ''I started chemotherapy...'' - ''My ordeal combined with the COVID-19 pandemic has been surreal.'' - ''I have had great support from my family and friends.''
**Conclusion**
Based on the analysis, the narrative events sequence (NES) includes receiving treatment and survivorship, and the narrator perspective (NP) is that of a breast cancer survivor.
**JSON Output**

```
{
  "NES": "3,4",
  "NP": "1"
}
```

This output captures the narrative events related to treatment and survivorship, and identifies the narrator as the breast cancer survivor herself.

**Agent 2:** ### Analysis of the Provided Text
**Narrative Event(s) Sequence (NES)**
The text describes several key events in Darlene Langley's breast cancer journey:
1. **Treatment**: The text mentions the start of chemotherapy on February 10, 2020, and the upcoming 25 days of radiation. This falls under ''Receiving treatment'' and ''Treatment milestone or completion.'' - ''I started chemotherapy on February 10, 2020...'' - ''After that I will have 25 days of radiation.'' - ''Reconstruction will begin six months after that.''

   2. **Survivorship**: The text also touches on the emotional and social aspects of her journey, including the support from family and friends, which is part of the survivorship experience.
      - ''My ordeal combined with the COVID-19 pandemic has been surreal.''
      - ''But through it all, I have had great support from my family and friends.''

Thus, the narrative events sequence includes: - Treatment (Receiving treatment, Treatment milestone or completion) - Survivorship
**Narrator Perspective (NP)**
The narrator in this post is Darlene Langley, a breast cancer survivor, as indicated by the direct quote and the attribution at the end of the post. - ''I started chemotherapy...'' - ''My ordeal combined with the COVID-19 pandemic has been surreal.'' - ''I have had great support from my family and friends.''
**Discussion of Other Social Scientists' Responses**
One social scientist suggested that the narrative events sequence (NES) should only include ''Treatment'' and not ''Survivorship.'' However, the emotional and social aspects of Darlene's journey, including the support from family and friends, are significant elements of survivorship. Therefore, I maintain that both ''Treatment'' and ''Survivorship'' should be included in the NES.
**Conclusion**
Based on the analysis and the discussion, the JSON output would be:

```
{
  "NES": "3,4",
  "NP": "1"
}
```

This captures the narrative events related to treatment and survivorship and identifies the narrator as the breast cancer survivor herself.
Judge: Same Agreement: True

TEXT: 4. ... (other disagreement text)

**\*\*\*\*\*\*\*\*\*\*Update codebook phase beginning \*\*\*\*\*\*\*\*\*\***

**Agent 1:** Based on the coding and discussion results, the original CODEBOOK appears to adequately fit the current examples. Each rule in the CODEBOOK has been applied at least once, and the rules are unique with minimal overlap. Therefore, there is no need to revise the CODEBOOK. Here is the original CODEBOOK for reference:

---

**CODEBOOK:**
**Codebook for Coders (July 1, 2021)**
**Narrative Elements in Facebook Posts by Breast Cancer Organizations**
**Narrative Event(s) related to breast cancer (plot)** [Code the occurrences of all events in the post, such as ''3'', ''2,4'', or ''2,4,5''] 1. **Prevention** 2. **Detection and/or diagnosis** 3. **Treatment** – Receiving treatment (e.g., getting the IV chemo, lying in the hospital bed) – Treatment effects (e.g., bald head, flat chest, wearing a head wrap) – Treatment milestone or completion (e.g., ringing the chemo bell, showing radiation therapy completion certificate). 4. **Survivorship** – includes complete remission/cancer free; recurrence; a second cancer; and death. 5. **Fundraising, any prosocial or philanthropic activities**
**Narrator perspective (narrator_perspective)** (evaluated based on the original post; choose one) [Narrator is the person telling the story. When coding, prioritize a perspective that is NOT the breast cancer organization.] 1. **Breast cancer survivor** 2. **Breast cancer survivor's family or friends** 3. **Mixed (i.e., survivor + family or friends)** 4. **Journalists/news media** 5. **Breast cancer organization**

---

This CODEBOOK has been applied consistently across the examples, and it covers all the necessary narrative elements and perspectives without redundancy. Therefore, no changes are needed.

**Agent 2:** Based on the coding and discussion results, the original CODEBOOK appears to adequately fit the current examples. Each rule in the CODEBOOK has been applied at least once, and the rules are unique with minimal overlap. Therefore, there is no need to revise the CODEBOOK. Here is the original CODEBOOK:
**CODEBOOK:**
**Codebook for Coders (July 1, 2021)**
**Narrative Elements in Facebook Posts by Breast Cancer Organizations**

- **Narrative Event(s) related to breast cancer (plot)**
    - [Code the occurrences of all events in the post, such as ''3'', ''2,4'', or ''2,4,5'']
    1. Prevention
    2. Detection and/or diagnosis
    3. Treatment
        - Receiving treatment (e.g., getting the IV chemo, lying in the hospital bed)
        - Treatment effects (e.g., bald head, flat chest, wearing a head wrap)
        - Treatment milestone or completion (e.g., ringing the chemo bell, showing radiation therapy completion certificate).
    4. Survivorship - includes complete remission/cancer free; recurrence; a second cancer; and death.
    5. Fundraising, any prosocial or philanthropic activities
- **Narrator perspective (narrator_perspective) (evaluated based on the original post; choose one)**
    - [Narrator is the person telling the story. When coding, prioritize a perspective that is NOT the breast cancer organization.]
    1. Breast cancer survivor
    2. Breast cancer survivor's family or friends
    3. Mixed (i.e., survivor + family or friends)
    4. Journalists/news media
    5. Breast cancer organization

This version of the CODEBOOK covers all cases and patterns in the examples provided, and each rule has been applied appropriately. Therefore, no changes are necessary.
Mediator: ### Summary of Opinions
Both social scientists agree that the original CODEBOOK is adequate and does not require any revisions. They both emphasize that:

1. The original CODEBOOK has been applied consistently across the examples.
2. Each rule in the CODEBOOK has been applied at least once.
3. The rules are unique with minimal overlap.
4. The CODEBOOK covers all necessary narrative elements and perspectives without redundancy.

**Updated CODEBOOK**
Based on the consensus that no changes are needed, here is the original CODEBOOK for reference:

---

**CODEBOOK:**
**Codebook for Coders (July 1, 2021)**
**Narrative Elements in Facebook Posts by Breast Cancer Organizations**
**Narrative Event(s) related to breast cancer (plot)** [Code the occurrences of all events in the post, such as ''3'', ''2,4'', or ''2,4,5''] 1. **Prevention** 2. **Detection and/or diagnosis** 3. **Treatment** – Receiving treatment (e.g., getting the IV chemo, lying in the hospital bed) - Treatment effects (e.g., bald head, flat chest, wearing a head wrap) - Treatment milestone or completion (e.g., ringing the chemo bell, showing radiation therapy completion certificate). 4. **Survivorship** – includes complete remission/cancer free; recurrence; a second cancer; and death. 5. **Fundraising, any prosocial or philanthropic activities**
**Narrator perspective (narrator_perspective)** (evaluated based on the original post; choose one) [Narrator is the person telling the story. When coding, prioritize a perspective that is NOT the breast cancer organization.] 1. **Breast cancer survivor** 2. **Breast cancer survivor's family or friends** 3. **Mixed (i.e., survivor + family or friends)** 4. **Journalists/news media** 5. **Breast cancer organization**

---

**Confirmation**
Dear Social Scientists,
Based on your feedback, the consensus is that the original CODEBOOK is adequate and does not require any revisions. Here is the original CODEBOOK for your reference. Please confirm if you agree with this conclusion and the provided CODEBOOK. If there are any further suggestions or concerns, kindly let me know.
Thank you for your collaboration.
Best regards, [Your Name]

**Agent 1:** ### Confirmation
Dear Social Scientists,
Based on your feedback, the consensus is that the original CODEBOOK is adequate and does not require any revisions. Here is the original CODEBOOK for your reference:

---

**CODEBOOK:**
**Codebook for Coders (July 1, 2021)**
**Narrative Elements in Facebook Posts by Breast Cancer Organizations**
**Narrative Event(s) related to breast cancer (plot)** [Code the occurrences of all events in the post, such as ''3'', ''2,4'', or ''2,4,5''] 1. **Prevention** 2. **Detection and/or diagnosis** 3. **Treatment** – Receiving treatment (e.g., getting the IV chemo, lying in the hospital bed) - Treatment effects (e.g., bald head, flat chest, wearing a head wrap) - Treatment milestone or completion (e.g., ringing the chemo bell, showing radiation therapy completion certificate). 4. **Survivorship** – includes complete remission/cancer free; recurrence; a second cancer; and death. 5. **Fundraising, any prosocial or philanthropic activities**
**Narrator perspective (narrator_perspective)** (evaluated based on the original post; choose one) [Narrator is the person telling the story. When coding, prioritize a perspective that is NOT the breast cancer organization.] 1. **Breast cancer survivor**

2. **Breast cancer survivor's family or friends** 3. **Mixed (i.e., survivor + family or friends)** 4. **Journalists/news media** 5. **Breast cancer organization**

---

Please confirm if you agree with this conclusion and the provided CODEBOOK. If there are any further suggestions or concerns, kindly let me know.
Thank you for your collaboration.
Best regards, [Your Name]

**Agent 2:** ### Confirmation
Dear Social Scientists,
Thank you for your thorough analysis and feedback. Based on our discussions and the consistent application of the original CODEBOOK across various examples, we have concluded that the original CODEBOOK is adequate and does not require any revisions. Here is the original CODEBOOK for your reference:

---

**CODEBOOK:**
**Codebook for Coders (July 1, 2021)**
**Narrative Elements in Facebook Posts by Breast Cancer Organizations**
**Narrative Event(s) related to breast cancer (plot)** [Code the occurrences of all events in the post, such as ''3'', ''2,4'', or ''2,4,5''] 1. **Prevention** 2. **Detection and/or diagnosis** 3. **Treatment** - Receiving treatment (e.g., getting the IV chemo, lying in the hospital bed) - Treatment effects (e.g., bald head, flat chest, wearing a head wrap) - Treatment milestone or completion (e.g., ringing the chemo bell, showing radiation therapy completion certificate). 4. **Survivorship** - includes complete remission/cancer free; recurrence; a second cancer; and death. 5. **Fundraising, any prosocial or philanthropic activities**
**Narrator perspective (narrator_perspective)** (evaluated based on the original post; choose one) [Narrator is the person telling the story. When coding, prioritize a perspective that is NOT the breast cancer organization.] 1. **Breast cancer survivor** 2. **Breast cancer survivor's family or friends** 3. **Mixed (i.e., survivor + family or friends)** 4. **Journalists/news media** 5. **Breast cancer organization**

---

Please confirm if you agree with this conclusion and the provided CODEBOOK. If there are any further suggestions or concerns, kindly let me know.
Thank you for your collaboration.
Best regards, [Your Name]
Judge: Same Agreement: True

