# OpenReview forum: "SCALE: Augmenting Content Analysis via LLM Agents and AI-Human Collaboration"
_ICLR.cc/2025/Conference — ICLR 2025 Conference Withdrawn Submission_

### Official Review · Reviewer_yvqB · 2024-10-29

**Soundness:** 3
**Presentation:** 3
**Contribution:** 2
**Rating:** 5
**Confidence:** 3

**Summary:**

The paper presents SCALE, an interesting multi-agent framework designed to simulate and augment the content analysis process using LLMs and AI-human collaboration. Automating key phases of content analysis, including text coding, inter-agent discussion, and codebook evolution could reduce the time, human resources, and costs traditionally required for content analysis. It also incorporates human intervention to mitigate algorithmic bias and improve contextual sensitivity. The paper suggests that SCALE could transform social science research by providing an efficient tool for analyzing large volumes of data.

**Strengths:**

1. The idea of using LLM agents and ai-human collaboration for content analysis is interesting.
2. The paper is easy to follow. For example, Fig. 2 is pretty detailed to explain the overall workflow of SCALE framework.
3. The paper could attract a large number of audience interested in using LLM agents to simulate social science research.

**Weaknesses:**

1. In the introduction, the authors mention that one of the drawbacks of using humans experts is the time and labor cost. The analysis of the proposed framework would benefit significantly if there is any analysis in terms of time/cost spent by humans for annotation vs LLMs.
2. I think Section 5.3 SUPERIOR PERFORMANCE OF SCALE should emphasize the overall quality of the whole framework instead of the single coding accuracy. The classification task is relatively trivial for LLMs.
3. Human evaluation (or detailed results) might be needed to assess the overall quality of using LLM agents to simulate content analysis beyond Codebook Update Phase.

**Questions:**

As weaknesses.

---

### Official Review · Reviewer_GA7q · 2024-10-31

**Soundness:** 2
**Presentation:** 2
**Contribution:** 3
**Rating:** 5
**Confidence:** 5

**Summary:**

The authors propose a framework for automatic content analysis of unstructured data with LLMs called SCALE. The framework includes the steps of automated text coding, inter-agent discussion and dynamic codebook updates, while also  allowing for human interventions. The goal is to develop a tool for social scientists that is able to support the process of content analysis at a large scale.

**Strengths:**

The paper has a clear strucure and demonstrates strengths and limitations of the method through several experiments. The topic is very relevant for the Social Science.

**Weaknesses:**

The central Figure (Figure 2) is not that easy to understand. It should be self-explanatory by reading the caption. It would be very helpful to know which dataset is taken as an example here. Unfortunately, the paper does not always read fluently. Sometimes articles are missing and there are some grammatical errors. Some technical details are missing, such as the implementation of the chain of thought and tree of thought baselines (or at least the references are missing - see below). Also the formula for the used accuracy measure should be written out (in my opinion).
The human intervention experiment is not really explained well. How much did the humans intervene? Is there a certain number of rounds? Is it the same setup as in the previous experiments?
Overall the idea of the framework and the process of inter-agent discussion for automated content analysis is good, but some important details are missing. It is also not clear from the paper how much manual effort is required to apply the whole framework. What are the necessary steps (e.g. developing personas, a first version for a codebook..)?
As the authors note at the end the inter-agent discussion introduces significant computational overhead. This leaves the question how practical the framework is.

Missing references:
- Chain of thought prompting as introduced by Wei et al (2022): Wei, Jason, et al. Chain-of-thought prompting elicits reasoning in large language models. Advances in neural information processing systems, 2022, 35. Jg., S. 24824-24837.
- Tree of thougt prompting: Yao, S., Yu, D., Zhao, J., Shafran, I., Griffiths, T., Cao, Y., & Narasimhan, K. (2024). Tree of thoughts: Deliberate problem solving with large language models. Advances in Neural Information Processing Systems, 36.
- Self Consitency: Wang, X., Wei, J., Schuurmans, D., Le, Q. V., Chi, E. H., Narang, S., ... & Zhou, D. Self-Consistency Improves Chain of Thought Reasoning in Language Models. In The Eleventh International Conference on Learning Representations.


Small errors:
* Line 088 should probably have a period at the end of "Human Intervention" to be consistent with the other items.
* line 190 is missing a period before (c)
* line 208: it should be N personas P, which ..., are derived from.. s
* The abbreviation NES is not introduced in the text
* line 241: "a K-round discussion" instead of "an K-round.."
* Line 320: It would be very nice if the Hamming loss was explicitly written out here in the formula.
* Line 461: lLM > LLM

**Questions:**

- The process of content analysis is always subjective isn't it? How does the method reduce subjectivity and is that even the goal?
- The appendix provides an overview of the different prompts associated with the different steps. How much manual effort is involved when applying the framework? Is the codebook really updated automatically or do the researchers have to manually extract codebook changes and copy them into their codebook?
- The framework is designed to iteratively update a codebook and use it as base for the coding. There are labels for each dataset. Were these labels the starting point for the coding task? How exactly were the experiments conducted? Did you only evaluate the coding step or did the experiments include the development of a codebook for each dataset?
- Why did you conduct the first experiments with only 2 agents?
- Are all tasks used for the experiments multi label tasks?
- Does the average accuracy of 0.7 refer to all models? You could also add a new column where you could plot the average.
- How much prompt engineering was involved in the process of building the framework? How did you come up with the different prompts? Do the results depend much on the wording of the prompts?

---

### Official Review · Reviewer_ViW6 · 2024-11-02

**Soundness:** 1
**Presentation:** 2
**Contribution:** 2
**Rating:** 3
**Confidence:** 4

**Summary:**

This paper proposes SCALE, a multi-agent framework to simulate content analysis using LLMs. The overall idea is to incorporate different phases of content analysis, such as text coding, inter-agent discussion, and codebook updating, in a comprehensive framework carried out by LLMs as a multi-step process. Additionally, the authors allow the framework for human intervention, to enhance AI-human expert collaboration. The SCALE framework was tested on five real-world datasets, spanning seven multi-class and multi-label classification tasks.

**Strengths:**

The core idea of this manuscript to leverage LLMs for augmenting content analysis is interesting, as can lead to improvements in capabilities (as the LLMs' intrinsic word model is rather rich and varied) and scalability (e.g., by alleviating the human burden in annotating large-scale content).

**Weaknesses:**

Despite focusing on an interesting and promising idea, this manuscript presents different criticalities, as follows:
- There is much emphasis on the capability of SCALE to incorporate domain knowledge of social science, yet this process seems limited to one (of many possible) prompting strategies, undermining the robustness and technical depth of the proposed framework.
- The experimental setup is not appropriate, as there is no comparison with baseline models (e.g., ML-based ones for sentiment analysis). Indeed, it is just confined to testing different prompting strategies, with two commercial models (i.e., GPT-4O and 4O-mini). Similarly, some experimental choices (e.g., the very low number of agents despite the sensitivity results) are not adequately motivated.
- The experimental results turn out to be particularly weak for 3 out of 7 tasks, with very low coding accuracies. Also, some additional quantitative measures (e.g., inter-agent agreement) would be beneficial for a better understanding of how SCALE handles the annotation processes.
- Despite aiming at fostering better human-AI interaction in content analysis, as well as strong capabilities, there is no human qualitative evaluation of the SCALE's capabilities. This would be needed to further validate the helpfulness of the proposed framework.
- The entire study relies solely on the GPT family of models. Experimenting with other (e.g., open) models would be beneficial for a broader applicability and adoption of the proposed framework.
- There are no technical details on the agents deployment and interaction. This is a key aspect for multi-agent systems, and should be stated in the manuscript to also foster reproducibility. Similar considerations hold for the human-in-the-loop setting.
- To properly validate how SCALE complements humans, there should be some more emphasis on the patterns occurring within it, and critical analysis on how the different phases differ or resemble humans. For instance, for RQ2, certain datasets see limited to no improvement after agents' discussion, why?

**Questions:**

- How do the authors ensure that SCALE is grounded in social science knowledge beyond simple prompting?
- Similarly, the authors claim (row 233) that agents [...]do not rely on external knowledge or data beyond what is provided in the codebook [...]. How do they ensure that agents do not leverage their own knowledge/biases in conducting content analysis, going beyond the received guidelines?
- Do the authors experiment with different initializations for the agents? That is, what is the effect of specifying agents' instruction, gender, and experience within prompts?
- As hallucinations are likely to occur with LLMs, how do the authors handle them?
- What is the default behavior when agents do not reach agreement within k iterations?
- As the temperature is not enough to reduce randomness in LLMs, which values did the authors use for top_p and top_k?

**Details Of Ethics Concerns:**

It appears that some names used as example scenarios (see row 237) actually exist and refer to real-life situations (as confirmed by a simple web search). In my opinion, these can and should be omitted (e.g., by replacing them with placeholder nicknames).

---

### Official Review · Reviewer_G8mV · 2024-11-04

**Soundness:** 2
**Presentation:** 2
**Contribution:** 1
**Rating:** 3
**Confidence:** 5

**Summary:**

The authors propose SCALE, which is a tool to perform content analysis in social science. By automating text coding and facilitating multi-agent discussion, SCALE approximates human judgment in text annotation tasks. The framework integrates AI-human collaboration, which mitigates algorithmic bias. The paper evaluates SCALE on diverse social science datasets, showing its effectiveness in improving large-scale content analysis.

**Strengths:**

SCALE introduces a multi-agentic approach to content analysis, enhancing scalability and efficiency by reducing the human resources required to make large-scale, high-quality analysis feasible. SCALE demonstrates high flexibility, adapting across multiple datasets without modifications. The paper might have a contribution to social sciences by enabling large-scale analysis traditionally constrained by labor-intensive methods.

**Weaknesses:**

One big concern about the paper is that it does not provide prior benchmarks. The datasets used by the authors are not very commonly used in this literature. I recommend that the authors use at least one dataset with prior benchmarks on multi-label classification (e.g., COCO, Peskine et al. 2023) or apply prior methodologies of multi-label classification on your datasets. How does the plain-vanilla GPT-4o perform on your dataset without including multiple agents?

It is well-known that agentic approaches improve LLMs’ performance. However, the approaches typically require more computational resources and time. It would be helpful if the authors could include each ablation's cost and processing time. The authors acknowledge this issue in Section 6, but it will be helpful for the readers to see the informational gain of this framework along with computational requirements.

In Section 5.4.3, the authors might want to include some desired codebook structures in their prompt. They could add layer of agents that review the final product by including several instructions, e.g., examining whether there are overlapping categories by providing some theory backgrounds. They might even try fine-tuning the LLMs using some domain knowledge instead of using the plain-vanilla versions.

Missing citations: Several works have already explored how the discussion among LLM agents can improve overall performance. For example, see Chan et al. (2023) and Kim et al. (2024). I’m surprised that the authors do not acknowledge any of these studies. At a high level, what this paper shows is similar to the value of a multi-agentic approach in classification problems.

References
Peskine, Youri, et al. "Definitions Matter: Guiding GPT for Multi-label Classification." Findings of the Association for Computational Linguistics: EMNLP 2023. 2023.
Chan, Chi-Min, et al. "Chateval: Towards better llm-based evaluators through multi-agent debate." arXiv preprint arXiv:2308.07201 (2023).
Kim, Alex, Keonwoo Kim, and Sangwon Yoon. "DEBATE: Devil's Advocate-Based Assessment and Text Evaluation." arXiv preprint arXiv:2405.09935 (2024).

Minor Comments
1)	You list five contributions but say the contributions are fourfold on page 2.
2)	Why are some datasets benefiting heavily from discussions while others are not (Figure 4)? It would be helpful to include some insights on where the discussions will likely improve the model performance more and why.
3)	In Table 3, it is concerning that you achieve the highest accuracy when human intervention is frequently made, and the LLM strictly follows it. Doesn’t this suggest that human interventions are required and LLMs alone cannot perform the task reliably?

**Questions:**

See "Weaknesses".

---

### Note · Authors · 2025-01-19

I have read and agree with the venue's withdrawal policy on behalf of myself and my co-authors.